

# Estimation of aerosol and cloud radiative heating rate in tropical stratosphere using radiative kernel method

Jie Gao[1,2], Yi Huang[3], Jonathon S. Wright[1], Ke Li[4], Tao Geng[2], Qiurun Yu[3]

[1]Department of Earth System Science, Ministry of Education Key Laboratory for Earth System Modeling, Tsinghua University, Beijing China
[2]Laoshan Laboratory, Qingdao, China
[3]Department of Atmospheric and Oceanic Science, McGill University, Montreal, QC, Canada
[4]Jiangsu Key Laboratory of Atmospheric Environment Monitoring and Pollution Control, Jiangsu Collaborative Innovation Centre of Atmospheric Environment and Equipment Technology, School of Environmental Science and Engineering, Nanjing University of Information Science and Technology, Nanjing, China

*Correspondence to:* Jie Gao (gaoj17@tsinghua.org.cn)

**Abstract.** A layer of aerosols has been identified in the upper troposphere and lower stratosphere above the Asian summer monsoon region, which is referred to as the Asian Tropopause Aerosol Layer (ATAL). This layer is fed by atmospheric pollutants over South and East Asia lifted to the upper troposphere by deep convection in summer. The radiative effects of this aerosol layer change local temperature, influence thermodynamic stability, and modulate the efficiency of air mass vertical transport near the tropopause. However, quantitative understanding of these effects is still very poor. To estimate aerosol radiative effects in the high atmosphere, a set of radiative kernels is constructed for the tropical upper troposphere and stratosphere to reduce the computational expense of decomposing the different contributions of atmospheric components to anomalies in radiative fluxes. The prototype aerosol kernels in this work are among the first to target vertically resolved heating rates, motivated by the linearity and separability of scattering and absorbing aerosol effects in ATAL. Observationally-derived lower boundary conditions and satellite observations of cloud ice within the upper troposphere and stratosphere are included and simplified in our Tropical Upper Troposphere-Stratosphere Model (TUTSM). Separate sets of kernels are derived and tested for the effects of absorbing aerosols, scattering aerosols, and cloud ice particles on both shortwave (solar) and longwave (thermal) radiative fluxes and heating rates. The results indicate that the kernels we calculated can well reproduce the aerosol radiative effects in ATAL, and these aerosol kernels are also expected to simulate radiative effects of biomass burning and volcanic eruption above troposphere. It has been proved this approach substantially reduces computational expense while achieving good consistency with direct radiative transfer model calculations. It can be applied to models that do not require high precision but have requirements for computing speed and storage space.

## 1 Introduction

As one of the most uncertain and complicated factors in atmospheric simulation and climate projection, aerosol plays an important role in atmospheric radiation budget (Kuniyal and Guleria, 2019; IPCC, 2021). In order to analyze heat balance and



temperature changes, researchers often need to simulate the radiation effects of aerosols (Brühl et al., 2018; Ge et al., 2022). A radiative kernel approach (Soden et al., 2008; Shell et al., 2008; Zelinka et al., 2012; Sanderson and Shell, 2012; Huang et al., 2017) could help to facilitate sensitivity studies on how direct and indirect radiative effect of aerosols.

Soden and Held (2006) defined a radiative kernel method, which is the expected change in radiative flux given a standard change in a radiatively relevant constituent or property. Radiative kernels essentially present the leading term in a Taylor series expansion of the radiative response to a perturbation from a given reference state. They have been demonstrated to be useful tools for understanding the sensitivity of the Earth's radiative balance to changes in atmospheric composition and other radiative forcing factors, for example aerosols. These kernels consist of precomputed sensitivities that relate changes in

atmospheric constituents to changes in radiative fluxes. When the radiatively active factors are independent, radiative kernels make it possible to separate the total climate feedback into distinct components (e.g., Zelinka et al., 2020). This decomposition approach provides a powerful framework for identifying the spatio-temporal characteristics of climate feedbacks and diagnosing the reasons for intermodel discrepancies in feedback strength (Soden et al., 2008). With these usages, radiative kernels are often used to calculate climate feedbacks in coupled model simulations, under the assumption that the radiative

response to changes in the variable of interest is linear (Sanderson et al., 2010; Jonko et al., 2012). When aggregated globally, these radiative effects of different factors can be approximated by radiative kernel functions with an accuracy of ±5% at greatly reduced computational costs relative to direct radiative transfer calculations (Soden and Held, 2006; Soden et al., 2008).

Thorsen et al. (2020) proposed a set of top of atmosphere (TOA) shortwave aerosol radiative kernels to reduce uncertainty in remote sensing calculations. They calculated kernels based on aerosol extinction coefficient, single scattering albedo, and

asymmetry factor, under the assumption that aerosol direct radiative effects are linear and can be approximated without additional radiative transfer calculations. The aerosol radiative kernels are calculated by perturbing 1% of each variable to validate the linear assumption. After simplifying the vertical structure of aerosol scattering properties, they estimated the deviation of clear-sky aerosol radiative kernel to be $-0.06$ W·m$^{-2}$ globally, increasing to $-0.22$ W·m$^{-2}$ under all-sky conditions. The radiative kernel based on extinction coefficient was more than 30 times larger than those based on single scattering albedo

or asymmetry factor. Moreover, thin cloud layers were found to have larger influences on aerosol radiative effects than thick cloud layers. Aerosol kernels calculated from MERRA-2 were also compared with CERES (The Clouds and the Earth's Radiant Energy System) observations by Kramer et al. (2021). The results showed similar spatial patterns, with differences resulting mainly from different trends in AOD. These results establish the viability of aerosol radiative kernels for TOA fluxes under the linear assumption.

Previous researches of the radiative kernel have focused almost exclusively on TOA and surface radiative fluxes, lacking calculations of the aerosol radiation kernel in the upper troposphere and stratosphere, which is important in some regions. The upper troposphere and lower stratosphere above the Asian summer monsoon (ASM) contains higher concentrations of aerosols and other tropospheric pollutants compared with other regions at similar latitudes (Dethof et al., 1999; Randel and Park, 2006; Park et al., 2007; Brunamonti et al., 2018). Much of the air within this region can be traced back to the boundary layer over



South and East Asia (Bergman et al., 2013; Orbe et al., 2015; Vogel et al., 2016; Zhang et al., 2020), where abundant emissions of aerosols and aerosol precursors contribute to the formation of an Asian Tropopause Aerosol Layer (ATAL) (Neely et al., 2014; Yu et al., 2015; Vernier et al., 2018; Bian et al., 2020). As an important pathway for air mass exchange between the troposphere and stratosphere (Pan et al., 2016; Yan et al., 2019; Bian et al., 2020), the ASM contributes up to 10% of annual global mass transport upward across the tropopause (Ploeger et al., 2017; Shi et al., 2018; Yan et al., 2019). Due to the fact

that radiation heating rate have contribution to local temperature, estimating the radiation heating rate through upper troposphere to stratosphere helps to better understand the mass transport above ASM. Enhanced aerosol concentrations in the upper troposphere lower stratosphere can alter radiative heating at the tropopause level both directly through their interactions with radiative fluxes (Toohey et al., 2014; Vernier et al., 2015; Yu et al., 2015; Fadnavis et al., 2017, 2019) and indirectly through their interactions with clouds (Su et al., 2011; Dong et al., 2019; Fadnavis et al., 2019). However, the practical impacts

of these aerosol effects depend in large part on the composition and vertical structure of the aerosol layer (Gao et al., 2023), both of which are highly variable (Hanumanthu et al., 2020) and poorly constrained (Bian et al., 2020), which need to be estimated in a flexible method.

Assessments of troposphere-to-stratosphere mass exchange in ASM region are typically based on Lagrangian models driven by reanalysis products (Fueglistaler et al., 2005; Wright et al., 2011; Ploeger et al., 2017; Nützel et al., 2019; Yan et al., 2019).

However, even in reanalysis products that include interactive aerosols, cloud processes and cloud-aerosol interactions near the tropopause are not well resolved. As a pre-calculated coefficient, aerosol radiative kernels would permit rapid estimation of aerosol effects on radiative heating and fluxes under different distributions and compositions of the ATAL, and reduce dependence on the reanalysis products used to drive Lagrangian models. The main goal of this work is to construct a set of aerosol radiative kernel in the upper troposphere and stratosphere (UTS), which can quantify the radiative effects of the ATAL.

Extension of the kernel framework to vertically-resolved radiative fluxes and heating rates is one of the main novelties of this work, and it also requires extensive validation.

The included datasets for radiative and transfer model used in this paper are introduced in section 2. The development of the kernels, including key assumptions and feasibility testing, is described in section 3. The reference state, kernel distributions, and validation of both shortwave and longwave kernels are presented in section 4. The radiative kernels calculated in this work

are applied to the ATAL in section 5, followed by a summary and discussion of the results in section 6.

## 2 Data and Model

The kernels are calculated by radiative transfer model with reanalysis data as input. For feasibility testing, input data are taken exclusively from Modern-Era Retrospective Analysis for Research and Applications Version 2 (MERRA-2) for convenience. However, when calculating the prototype kernels for application, several variables are taken from Aura Microwave Limb

Sounder (MLS) observations and the ERA5 reanalysis to account for known biases in MERRA-2 ozone, water vapor, and



temperature tendency (Davis et al., 2017; Wright et al., 2020; Fujiwara et al., 2022). For the final calculations, we use aerosol concentrations from MERRA-2; water vapor, ozone, and ice water content from MLS; temperature and height from ERA5. All inputs to the radiative transfer model RRTMG (Rapid Radiative Transfer Model for General Circulation Models) are interpolated to the MLS vertical levels (Waters et al., 2006), which provide a finer resolution in the stratosphere relative to

MERRA-2. The longwave and shortwave radiative fluxes from CERES (Clouds and the Earth's Radiant Energy System) satellite data are also used for validation test. The datasets included in this work are listed in table 1.

**Table 1: Datasets used in RRTMG.**

| Aerosol | Atmospheric component | Validation |
|---|---|---|
| MERRA-2 (dust, organic carbon, black carbon, sulfate) | MLS (cloud ice, water vapor, ozone) ERA5 (height, temperature) | CERES (shortwave and longwave radiative flux) |

## 2.1 Reanalysis products

The MERRA-2 is produced by NASA (National Aeronautics and Space Administration) Global Modeling and Assimilation

Office (GMAO) during the satellite era (1980 - present) (Randles et al., 2017). The Goddard Chemistry, Aerosol, Radiation, and Transport (GOCART) model is used by MERRA-2 in simulating aerosols and making the integration between meteorological and aerosol observation data into the global assimilation system (Randles et al., 2017). Aerosols in MERRA-2 are assimilated by aerosol optical depth (AOD) from AVHRR (Advanced Very High Resolution Radiometer), MODIS (Moderate Resolution Imaging Spectroradiometer), MISR (Multi-Angle Imaging SpectroRadiometer) and AERONET

(Aerosol Robotic Network), and not directly coupled with cloud (Buchard et al., 2015, 2017). The AOD in MERRA-2 are only 2-dimensional, so the 3-dimensional aerosol mixing ratio with 14 species is chosen and converted to AOD with MERRA-2 lookup table in this work for the calculating of 3-dimensional radiative kernel. The MERRA-2 aerosols are interacted within radiation calculations, which are unique among current meteorological reanalysis (Fujiwara et al., 2022), that provide a way for the validation of radiative model output.

The Fifth Generation of ECMWF Atmospheric Reanalysis (ERA5) is a state-of-the-art global atmospheric reanalysis produced by the European Centre for Medium-Range Weather Forecasts (ECMWF). It provides a comprehensive view of the atmosphere by assimilating vast amounts of observations, including surface and satellite measurements, radiosonde soundings, and aircraft measurements (Bell et al., 2021). ERA5 compares favorably with other recent reanalyses, such as MERRA-2, ERA-Interim, and JRA-55 (Japanese 55-year Reanalysis) (Hersbatch et al., 2020), better representing the vertical structure of temperature,

ozone, and water vapor profiles near the tropopause relative to MERRA-2 (Fujiwara et al., 2022), and is therefore used to as the background state for kernel calculations.



## 2.2 Observations

The Microwave Limb Sounder (MLS) is an instrument onboard the NASA Aura satellite that uses passive microwave radiometry to measure the concentration of trace gases such as ozone, water vapor, and methane, as well as temperature and cloud ice (Waters et al., 2006). MLS data coverage extends from August 2004 to the present and spans the upper troposphere, stratosphere, and mesosphere. The original objective of the MLS instrument was to detect chlorine monoxide (ClO), which is emitted by industrial activity and can destroy ozone layer (Manney et al., 2020), but the products have become invaluable datasets for studying chemical and physical processes in the upper atmosphere.

The Clouds and the Earth's Radiant Energy System (CERES) instruments on the Terra and Aqua spacecraft provide measurements of global TOA fluxes since March 2000. Filtered broadband fluxes measured by CERES span the spectral range from 0.3 - 11.8 μm and include both shortwave (0.3 - 5 μm) and window (8.1 - 11.8 μm) regions. This study uses version 4.3 of the CERES synoptic 1° (SYN1deg), which uses calibrated TOA fluxes from geostationary satellites and model simulations to fill gaps in the diurnal cycle not measured by Terra or Aqua CERES (Doelling et al., 2016). The CERES SYN1Deg product includes observed and simulated TOA fluxes, simulated surface fluxes, and simulated fluxes at selected vertical levels, along with cloud properties and aerosols from MODIS. The data are provided on a 1°×1° horizontal grid at temporal resolutions ranging from hour to month.

## 2.3 Radiation model

For calculating radiative kernel with offline radiative transfer model, RRTMG is chosen. The RRTMG is a widely-used radiative transfer model that uses the correlated-k approach to calculate radiative fluxes and heating rates (Mlawer et al., 1997). Shortwave and longwave radiative transfer are calculated using two separate tools, RRTMG_SW and RRTMG_LW, with respective spectral ranges of $825 - 50000$ cm$^{-1}$ ($0.2 - 12.2$ μm for wavelength) and $10 - 3250$ cm$^{-1}$ ($3.08 - 1000$ μm for wavelength). The models take into account absorption, emission, and scattering of radiation by atmospheric gases, aerosols, and clouds, including water vapor, $CO_2$, ozone, $N_2O$, methane, and some common halocarbons (Mlawer et al., 1997).

RRTMG has been implemented in a variety of atmospheric models, including the GEOS-Chem chemical transport model (Heald et al., 2014), the WRF (The Weather Research and Forecasting Model) weather research and forecasting model (Skamarock and Klemp, 2008), and the Community Earth System Model (CESM) (Hurrell et al., 2013). Overall, RRTMG is a widely used and well validated radiative transfer model that has been widely implemented and has contributed significantly to our understanding of the Earth's energy balance and climate system. Input data required by RRTMG include basic conditions (such as solar zenith angle, insolation, surface emissivity, and temperature), concentrations of some atmospheric components (such as $CO_2$, $H_2O$, $O_3$, and $N_2O$), aerosol and cloud concentrations and optical properties.

To improve our ability to represent the radiative effects of aerosol-cloud interactions through the kernels, we describe cirrus cloud ice using an aerosol-type input file that specifies optical depth, single scattering albedo, and asymmetry factor. These



optical properties of cirrus cloud ice are parameterized following the equations suggested by Fu et al. (1996) and Fu et al. (1998).

## 3 Kernel feasibility and testing

### 3.1 Linearity assumptions

Radiative kernels can be regarded as sensitivity parameters or functions that encode the response of radiative fluxes to a standardized perturbation of a specific radiatively active component relative to a reference condition. Although nonlinear kernels can be constructed (Bani Shahabadi and Huang, 2014; Huang and Huang, 2021), most kernels are based on the assumption that the perturbations of interest can be linearized (Soden and Held, 2006; Soden et al., 2008; Shell et al., 2008; Kramer et al., 2019). Linear kernels have previously been used to represent the sensitivity of radiative fluxes to aerosol types and concentrations (Thorsen et al., 2020). Key motivations for using the linear assumption include simplicity and convenience, as well as the reduction of computational cost. Moreover, linear kernels have proven to be accurate across many applications. The aerosol radiative kernels proposed in this work are thus calculated based on the following equation (Huang et al., 2017):

$$k_x = \frac{\partial R}{\partial x}, \tag{1}$$

in which $k_x$ is the pre-computed kernel sensitivity parameter for the radiative component, $R$ is the radiative flux or radiative heating rate, $x$ represents aerosol optical depth (AOD) in this work. It should be noted that aerosol radiative kernels defined in this way only encode the direct radiative forcing due to aerosol, and do not include any corresponding variations in clouds or albedo. Kernels are used to estimate changes in radiative forcing relative to a reference state. The perturbed radiation can be calculated by the linear kernel as $k_x \Delta x$, where $\Delta x$ represents the difference in variable x relative to the reference state.

To get a rough estimation of the linearity of aerosol radiative effects, the sensitivity is tested to increase the total column aerosol optical depth in the core region of ATAL (20°N -25°N, 85°E-90°E), where aerosol concentration is large, incrementally from its average value to 10 times its average value. The corresponding variations in net flux at the TOA, net flux at the surface, and heating rate at the tropopause above ASM are shown for clear-sky conditions in Fig. 1 and all-sky conditions in Fig. S1. The solar zenith angle has a large impact on aerosol radiative effects on fluxes at the surface and TOA (Gao et al., 2023). We therefore evaluate shortwave aerosol radiative effects at eight times of day separated by 3-hour intervals. The hours listed in Fig. 1 and Fig. S1 are reported as UTC times, which are 6 hours earlier than local solar times (UTC+6). Under clear-sky conditions, radiative effects are strong only at 03UTC (overlapped by result in 09UTC), 06UTC, and 09UTC (9:00, 12:00, and 15:00 local time). However, the aerosol effects on TOA and surface radiative fluxes are not well represented by linear extrapolation when the radiative effects are large. Aerosol radiative flux on TOA and surface during mid-day hours with small solar zenith angles fit the linear model only when AOD is within a factor of two (in 03UTC and 09UTC) or four (in 06UTC) of its reference state. When AOD is increased beyond this threshold, the linear assumption results in large overestimates of the





radiative effect because the shading effect of aerosols is close to saturation. Similar results are found for all-sky TOA and surface fluxes, large AOD can even leads to negative surface flux (Fig. S1). However, aerosols effects on heating near the tropopause are well represented by a linear model throughout the range of AOD tested in these simulations, up to 10 times the average value.

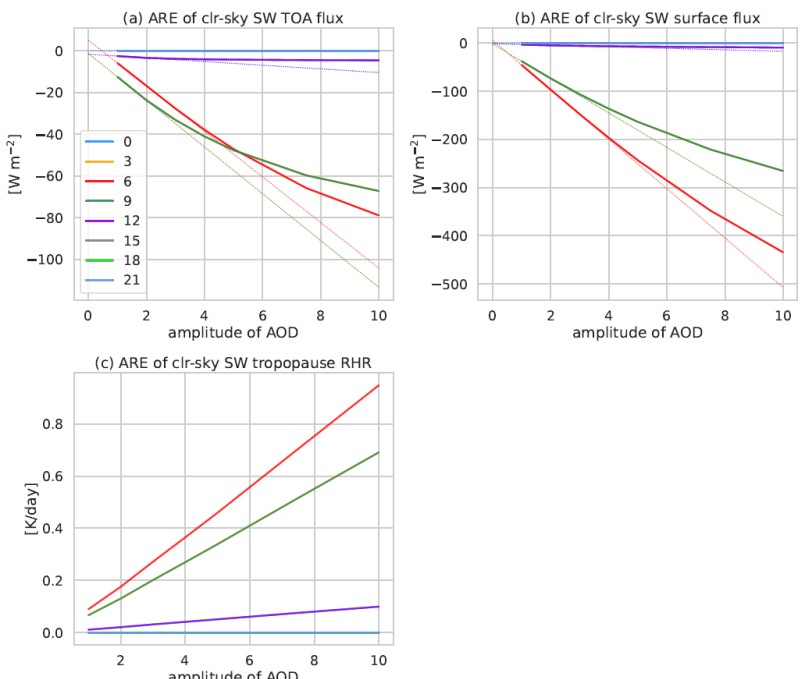

**Figure 1: Variations in clear-sky shortwave aerosol radiative effects based on different amplitudes of total AOD (100% - 1000%) at 22.5°N – 25°N, 87.5°E – 90°E for (a) TOA flux, (b) surface flux, and (c) heating rate at the tropopause (100 hPa) at eight times of day (time in UTC). Dotted lines show linear trends extrapolated from the 10% change between 100% and 110% of total AOD. Perturbations to AOD are uniform within the column.**

In summary, the linearity assumption breaks down for aerosol effects on surface and TOA fluxes when AOD perturbations exceed twice the reference value. Therefore, linear kernels are unsuitable for reconstructing these effects when perturbations are large. By contrast, linear kernels show excellent potential for representing aerosol effects on heating at the tropopause even when perturbations are very large.

### 3.2 Kernel calculation methodology

Having established the linearity of the aerosol effects, there remain many other parameters that can influence the accuracy of the kernels. We calculate shortwave clear-sky kernels under different aerosol settings to evaluate these sensitivities. The ASM core region (22.5°N – 25°N, 87.5°E – 90°E) is chosen as the reference. The atmospheric reference state is taken as the July 2020 average from MERRA-2 and the target state as July 2019.



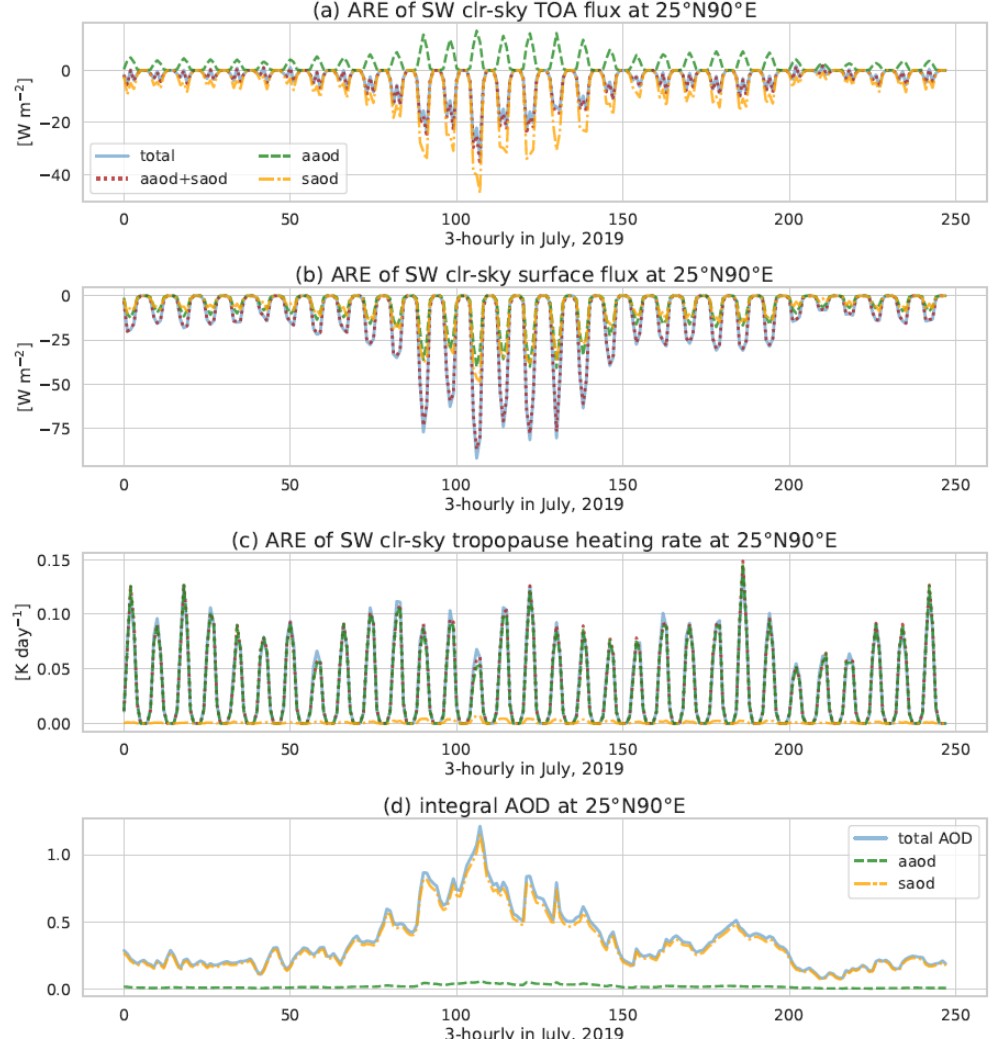

**Figure 2: The shortwave radiative effects of aerosol on (first row) TOA radiative flux, (second row) surface radiative flux, and (third row) heating rate at the tropopause attributed to absorbing aerosol (AAOD, green dashed lines), scattering aerosol (SAOD, orange dashed lines), absorbing and scattering aerosol (total, blue solid lines), and the sum of AAOD and SAOD radiative effects (red dotted lines). The fourth row shows time series of the vertically integrated AAOD, SAOD, and total AOD. All panels are 3-hourly data from July 2019 in the ASM core region.**

To separate the radiative effects of absorbing and scattering aerosols, Thorsen et al. (2020) proposed that aerosol direct radiative effects could be reconstructed using separate kernels for AOD, SSA, and asymmetry factor. However, SSA and asymmetry factor are less widely used than AOD and have much less influence. We therefore test a different approach, constructing kernels based on absorbing aerosol optical depth (AAOD) and scattering aerosol optical depth (SAOD). To test





the feasibility of linearly separating the AOD effect into AAOD and SAOD effects, changes in TOA radiative fluxes, surface fluxes, and tropopause heating associated with differences in AAOD, SAOD, and total AOD are plotted in Fig. 2. Although AAOD accounts for a small proportion of the total AOD, it contributes almost all of the aerosol effect on radiative heating at the tropopause and makes substantial contributions to the aerosol effects on TOA and surface fluxes as well. Changes in SAOD,

which accounts for 90% of total AOD, exert substantial effects on TOA and surface fluxes; however, these effects are similar in both sign and magnitude. SAOD has little impact on heating at the tropopause. The aerosol effect on downward TOA radiative flux represents competition between the positive effect of absorbing aerosol and the negative effect of scattering aerosol, like Fig. S1 (a). The sum of AAOD and SAOD radiative effects is in good agreement with the radiative effect of total AOD, indicating that it is feasible to reconstruct the total radiative effect from separate kernel representations of the AAOD

and SAOD effects.

To test and calculate aerosol radiative kernels, aerosols are perturbed by increasing concentrations by 10% at each level simultaneously. We change the time scale and independent variables to calculate kernel, therefore the parameters of fitting line in the scatter plot with original values on the x axis and kernel calculated values on the y axis are changed (e.g. Fig. 3 d, e, f), shown in Table 2. The closer the slope is to 1, the better the kernel method is. We first calculate monthly kernels of AOD,

but find they have a low consistency with direct model output. As shortwave radiation is largely influenced by insolation and solar zenith angle, kernels for each time step are needed, but that method can be easily influenced by specific features of the reference state. It is therefore necessary to refine the shortwave kernel calculation method, such as by using carefully selected time or spatial averages to set the reference state.

**Table 2: Summary of shortwave kernel method.**

| Time scale | Kernel variables | Samples in kernel calculation | Slope of TOA flux | Slope of surface flux | Slope of heating rate |
|---|---|---|---|---|---|
| monthly | AOD | 248 | 0.834 | 0.821 | 0.952 |
| monthly | AAOD, SAOD | 248 | 0.848 | 0.814 | 1.004 |
| monthly, diurnal cycle | AOD | 31 | 0.739 | 0.968 | 0.907 |
| monthly, diurnal cycle | AOD, SSA | 31 | 0.771 | 0.946 | 0.928 |
| monthly, diurnal cycle | AAOD, SAOD | 31 | 0.906 | 0.929 | 0.971 |

In Fig. 3, we apply monthly mean AAOD and SAOD kernels to calculate the total radiative effect. Each kernel is calculated as:





$$k_{\tau,t} = \overline{\left( \frac{R_{1.1\tau,t,\text{ref}} - R_{\tau,t,\text{ref}}}{\tau_{t,\text{ref}} \times 10\%} \right)}, \tag{2}$$

in which the subscript t indicates the 3-hour time step relative to 0UTC, and the subscript ref indicates the reference state (i.e.,

235 year 2011 - 2020). $\tau_{t,\text{ref}}$ means integrated aerosol optical depth, may be either AAOD or SAOD. $k_{\tau,t}$ is the radiative kernel for AOD, $R_{1.1\tau}$ is the radiative flux following a 10% increase in AOD, and $R_\tau$ is the radiative flux for the reference state AOD. The kernel for each time step is first calculated and then averaged monthly but diurnal variation remained. Noting that we use $\tau_{t,\text{ref}} \times 10\%$ in the denominator, the total radiative effect at the target time can be reconstructed as:

$$R_{\tau,t} = \left( \tau_{A,t} - \tau_{A,t,\text{ref}} \right) \times k_{A,t} + \overline{R_{A,\text{ref}}} + \left( \tau_{S,t} - \tau_{S,t,\text{ref}} \right) \times k_{S,t} + \overline{R_{S,\text{ref}}}, \tag{3}$$

in which $k_A$ and $k_S$ are the monthly AAOD and SAOD kernels, respectively, and $\overline{R_{A,\text{ref}}}$ and $\overline{R_{S,\text{ref}}}$ are the monthly averaged

AAOD and SAOD radiative effects for the reference state relative to no-aerosol condition.

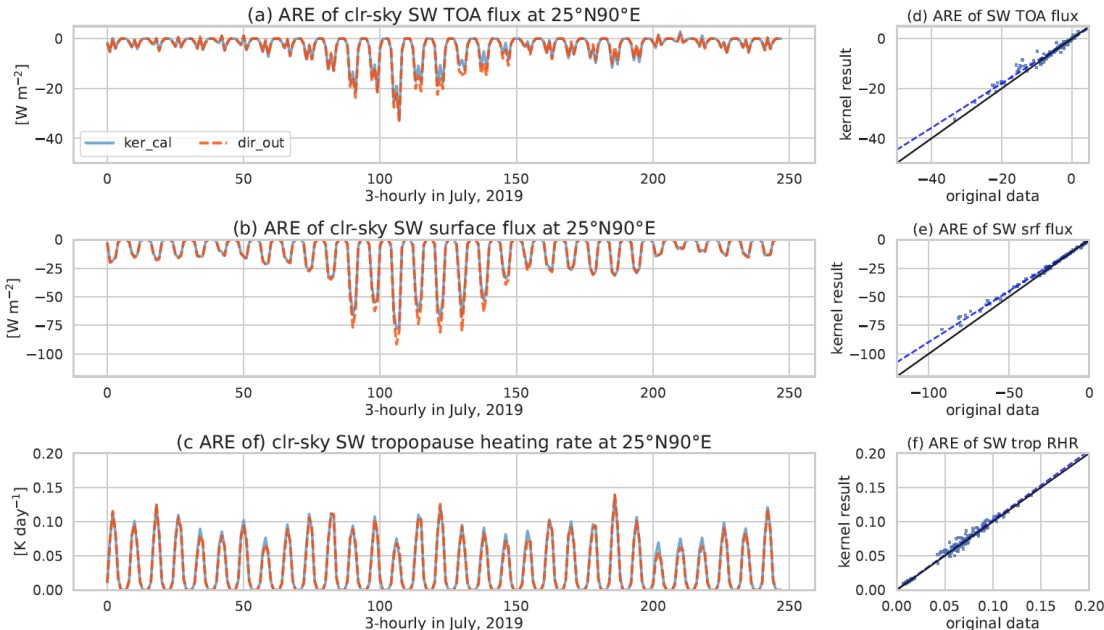

**Figure 3: The left column shows time series of the sum of absorbing and scattering aerosol radiative effects on (a) TOA radiative fluxes (positive upward), (b) surface radiative fluxes (positive downward), and (c) tropopause heating. Orange dashed lines are for ARE fluxes calculated directly from model outputs, blue solid lines are for ARE fluxes calculated**

**using diurnal resolved monthly aerosol kernels based on AAOD and SAOD as in Eq. 3. The right column shows scatter plots of direct (x-axis) and kernel-based (y-axis) aerosol radiative effects on (d) TOA fluxes, (e) surface fluxes, and (f) tropopause heating.**



The shortwave kernel and reference aerosol radiative effects are averaged across different days but retain variations associated with the diurnal cycle. The 3-hour resolution of the reference state thus yields 8 sets of kernels per month and the requirement that values during local night be manually set to zero. The agreement in radiative fluxes using this method is high, and the agreement in heating rate is the best, with the strong agreement indicated by a regression slope of 0.971 and a small root-mean-square error (RMSE) of 0.02. While for longwave aerosol kernel, scattering aerosol is not included in RRTMG, so only AAOD kernel is calculated.

## 4 Kernel in Tropical Upper Troposphere-Stratospheric Model (TUTSM)

### 4.1 Assumptions

In our upper troposphere stratosphere aerosol kernels, the reference state is based on monthly zonal-mean profiles of atmospheric temperature and trace gases. These choices are justified by relatively small zonal variations in the mean state and a weak sensitivity of aerosol effects to background conditions. As outlined in section 2, aerosol concentrations are from MERRA-2; temperature, height, and surface albedo are from ERA5; and ozone, water vapor, and cloud ice water content within the UTS are from Aura MLS. The monthly diurnal cycle is resolved at 3-hour intervals when computing the shortwave kernels. The diurnal cycle is not considered when computing the longwave kernels.

Kernels are constructed for latitude bands at 5° intervals in the tropics (30°S – 30°N), with 200 hPa as the lower boundary when running the radiative transfer model RRTMG. This tropical upper troposphere-stratospheric model (TUTSM) is schematic illustrated in Fig. 4. The troposphere is regarded as an entirety responsible only for setting the albedo and longwave upward radiative flux at the 200-hPa lower boundary (orange band and red arrows in Fig. 4). Cirrus cloud ice in the UTS (grey dots) is treated as a separate class of aerosol (brown dots).

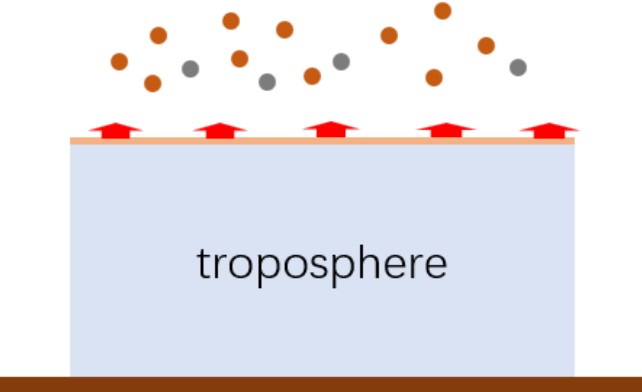

**Figure 4: Diagram of the UTS domain for which kernels are constructed. The troposphere (blue box) is considered only with respect to setting the lower boundary conditions at 200 hPa (orange band above troposphere). Red arrows indicate**



**upward fluxes of reflected shortwave or emitted longwave radiation from the troposphere. Orange and grey dots represent aerosols and cloud ice in the UTS, and the latter is also treated as aerosol when running the model.**

With these assumptions, it is necessary to define the 200 hPa albedo for shortwave radiation along with an effective tropospheric emission temperature based on the upward flux of longwave radiation at that level. The albedo at 200 hPa is calculated as the ratio of shortwave upward flux to downward flux at 200 hPa, written as:

$\quad \alpha = E_{\mathrm{SW,up,200hPa}}/E_{\mathrm{SW,down,200hPa}}$ , $\hspace{5cm}$ (4)

Because of the small optical depth for reflected shortwave radiation in the stratosphere, the albedo calculated at 200 hPa is almost the same as that calculated at the TOA, as shown in Fig. S2. So, the TOA albedo can be roughly used instead of 200 hPa albedo in applications. The emission temperature is calculated from longwave upward fluxes according to the Stefan–Boltzman Law:

$\quad T_e = \sqrt[4]{E_{\mathrm{LW,up,200hPa}}/\sigma}$ , $\hspace{5cm}$ (5)

where $T_e$ is the emission temperature, $E_{\mathrm{LW,up,200hPa}}$ is the upward longwave flux at 200 hPa, and $\sigma = 5.67 \times 10^{-8} \, \mathrm{W} \cdot \mathrm{m}^{-2} \cdot \mathrm{K}^{-2}$ is the Stefan–Boltzmann constant.

By analysing distributions of tropospheric albedo and longwave upward fluxes at 200 hPa within 30°S – 30°N (e.g. Fig. S3), we identify four representative reference state boundary conditions for the radiation simulations. Analysing joint plots at 5°
intervals within 30°S – 30°N for each month (not shown here), we finally select four representative points as listed in Table 3. The upper troposphere lower stratosphere aerosol kernels are based on these four scenarios, which broadly represent clear-sky, low cloud, middle cloud, and high cloud conditions in the underlying troposphere, the frequency of those scenarios decreases sequentially.

**Table 3: 200-hPa boundary conditions for the four representative tropospheric scenarios.**

| Albedo | Longwave upward flux | Emission temperature | Scenario |
|--------|---------------------|---------------------|----------|
| 0.1 | 300 | 269.7 | Clear-sky |
| 0.25 | 290 | 267.4 | Low cloud |
| 0.45 | 200 | 243.7 | Middle cloud |
| 0.65 | 140 | 222.9 | High cloud |

It should be noted that the shortwave radiation is largely influenced by time and latitude because of different insolation and solar zenith angle. However, the longwave upward radiation is independent on those factors. The radiative flux above 200 hPa is strongly positively correlated with boundary condition, which is emission temperature at 200 hPa for longwave. And the correlation coefficients calculated from CERES are all beyond 0.99 between 30°S - 30°N, which highlights the latitude




independence of UTS longwave radiation. Based on those analysis, different with shortwave, the longwave kernels are calculated without distinguishing between time and location.

In summary, the assumptions of the kernel calculations and applications include:

(1) The aerosol radiative effect varies linearly within a range of aerosol optical depth.

(2) Other radiatively active components like water vapor and ozone do not have large variations in UTS.

(3) The total radiative effect of aerosols at each level can be represented as a linear sum of radiative effects associated with absorbing aerosol optical depth (AAOD), scattering aerosol optical depth (SAOD), and cloud optical depth (COD).

(4) Cirrus cloud ice above the lower boundary (200 hPa) can be represented as aerosol for radiation calculation.

(5) Stratospheric components can be well represented by their zonal and monthly means.

(6) Tropospheric radiative effects can be represented as a boundary condition.

(7) The selected value pairs of 200 hPa albedo and emission temperature can adequately represent clear-sky, low cloud, middle cloud, and high cloud tropospheric scenarios.

(8) Time and location have no influence on longwave radiation above 200 hPa.

## 4.2 Validations

To validate the assumption of TUTSM, we compare the UTS simulation results with adjusted CERES fluxes at 200 hPa and TOA, as well as results from radiative transfer model simulations that include the troposphere. The radiative transfer simulations are conducted assuming a clear–clean troposphere (no aerosol and no cloud) to reduce the confounding influence of inhomogeneities in aerosols and clouds. To account for longitudinal differences in atmospheric components and their impacts on radiation, we extract the CERES radiative fluxes at the 10%, 50%, and 90% percentiles for the corresponding latitude and time of day.

Figure 5 and Fig. 6 show comparisons of 200 hPa upward and downward shortwave fluxes for all-sky conditions in CERES (grey lines), in whole-column simulations (purple lines), and the four UTS scenarios at three different local solar times. We choose the local solar times with the largest fluxes, between 9:00 to 15:00. During this period, the upward flux for simulations that resolve the clear-clean troposphere (purple lines) fall between the UTS simulations with albedo values of 0.1 (blue lines) and 0.45 (green lines) of stratospheric simulation. The CERES upward fluxes are also broadly consistent, generally falling within the range of albedo we defined. The highest albedo scenario (0.65) falls outside the 90th percentile, as may be expected given the relative rarity of deep convection. The 200 hPa albedo is smaller at 12:00, and the CERES data are mostly within 0.1 to 0.45, that means 0.65 is not needed for this time. Deviations of these simulations with CERES scenarios occur near




15°N – 30°N, which represents zonal-mean surface albedo being larger at those latitude. Our assumptions for the representative scenarios are therefore reasonable in the sense that these four reference states span the range of possible boundary conditions.

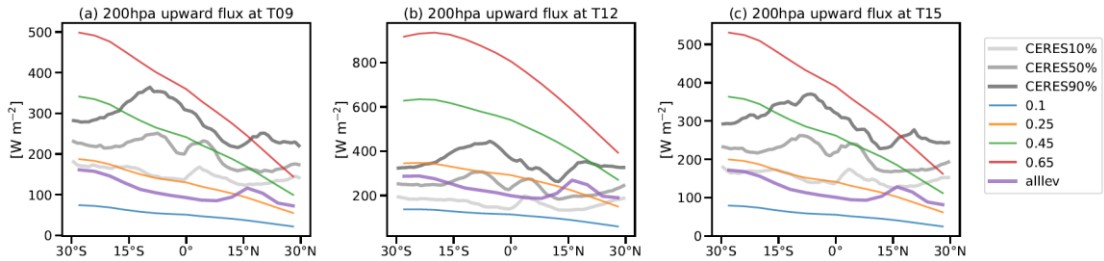

**Figure 5: Upward shortwave fluxes at (a) 9:00, (b) 12:00, and (c) 15:00 local solar time from all-sky UTS simulations (with AAOD, SAOD, and COD) for 200 hPa albedo equal to 0.1 (blue lines), 0.25 (yellow lines), 0.45 (green lines), and 0.65 (red lines). Purple lines show the 200 hPa upward flux from atmospheric simulations that resolve a clear–clean troposphere; grey lines are the 10th (light grey lines), 50th (moderate grey lines), and 90th (dark grey lines) percentiles of 200 hPa upward fluxes from CERES SYN1Deg for the corresponding latitude and approximate local solar time. The simulated results are based on monthly zonal-mean reference states for January 2011 – 2020. The CERES data are from January 2011 – 2020.**

Unlike the large differences in upward flux at 200 hPa between the different scenarios in upward flux, the tropospheric state has little impact on downward shortwave flux at 200 hPa. The total column simulation results are almost identical to the albedo = 0.1 TUTSM scenario, consistent with this scenario representing a clear–clean troposphere. The model simulations are within the range of CERES data at 12:00 but smaller than CERES-based fluxes at 9:00 and 15:00.

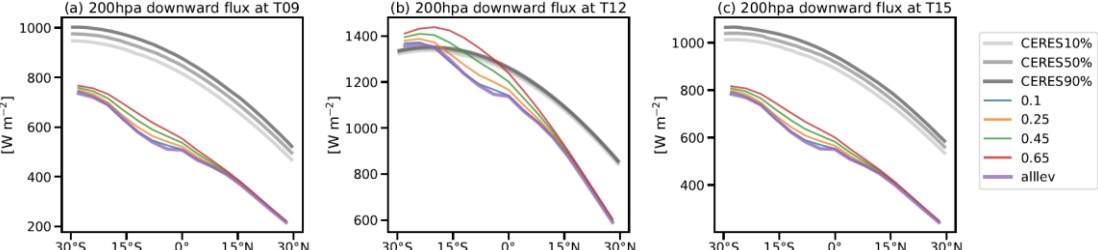

**Figure 6: Same as Fig. 5 but for downward flux.**

For validation of the longwave reference states, we ignore time of day and show both clear-sky and all-sky upward and downward longwave fluxes at 200 hPa. Results are shown in Fig. 7. As we use the same four 200 hPa effective emission temperatures at all latitudes, the set of reference longwave upward fluxes in the UTS simulations is the same for each latitude while the CERES-based and total column simulations show variations across latitudes. Under clear-sky conditions, the total column results and CERES data are close to the scenarios with 269.7 K and 267.4 K emission temperatures, consistent with the lack of clouds. For all-sky upward fluxes, these results fall mainly between the 267.4 K and 243.7 K emission temperature



scenarios, rarely consistent with deep convection (222.9 K emission temperature). For the all-sky downward flux, the CERES
fluxes are much smaller than the simulations between 30°S – 15°N. The simulations use zonal-mean MLS to define trace gases
and cloud fields for input to RRTMG_LW above 200 hPa. The use of zonal-mean cloud fields means that the simulations are
more representative of clear-sky than all-sky conditions.

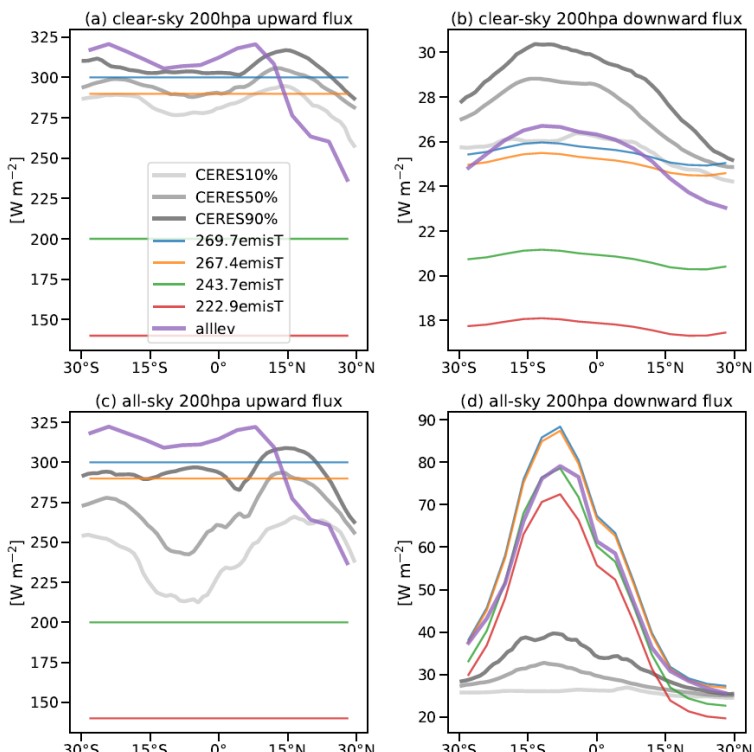

**Figure 7: (a) Upward longwave fluxes at 200 hPa for clear-sky and (c) all-sky conditions based on UTS simulations**
**with 200 hPa emission temperatures equal to 269.7 K (blue lines), 267.4 K (yellow lines), 243.7 K (green lines), and**
**222.9 K (red lines); whole-atmosphere simulations (including the troposphere; purple lines); and the 10th (light grey**
**lines), 50th (moderate grey lines), and 90th (dark grey lines) percentiles of CERES SYN1Deg data. (b, d) Same as left**
**but for downward longwave fluxes at 200 hPa. All results are for January 2011 – 2020.**

In conclusion, differences between the UTS-only simulations and CERES-based estimates or total atmosphere simulations are
355 generally small. Even in situations differences are not small, it meets expectations. These results therefore support the use of
200 hPa albedo (or planetary albedo, as they are almost the same in Fig. S2) and 200 hPa emission temperature as a
simplification of the tropospheric lower boundary condition. This simplification ensures reproducibility while greatly reducing
the number of computations required to produce a representative set of kernels. However, it also eliminates any band or
wavelength dependence of upward fluxes on conditions in the troposphere, and therefore may alter the simulated influences
of aerosols and trace gases and radiative fluxes at the TOA and radiative heating in the UTS.



### 4.3 Description of TUTSM radiation

#### 4.3.1 Shortwave radiation

Base on the structure and assumptions in section 4.1, we conduct radiative transfer simulations for UTS aerosols and cloud ice under the four representative scenarios (Table 3). We then use the results of these simulations to construct aerosol and cloud kernels for UTS region. Distributions of optical depth, shortwave net flux, and shortwave heating at 12:00 UTC in August are shown in Fig. S4 under the scenario corresponding to a cloud-free troposphere (i.e., 200 hPa albedo of 0.1). In August, deep convection associated with the ASM and the northward-shifted ITCZ (Intertropical Convergence Zone) injects relatively large amounts of aerosol and cloud ice in the northern part of the tropical UTS. The absorbing component of this aerosol leads to negative effects on shortwave net flux at altitudes with high concentrations and positive effects on net flux above these aerosol layers. By contrast, the effects of scattering aerosols and cloud ice on shortwave net flux have almost no vertical variations. Scattering aerosol produces positive upward perturbations in shortwave flux above layers with high concentrations of scattering aerosols while reducing the downward flux below (not shown). This compensation between changes in upward and downward flux induced by scattering aerosols results in a near-constant vertical distribution. The effects of cirrus cloud ice are similar. Compared with the strong radiative heating effect of absorbing aerosol, the effects of scattering aerosols on radiative heating can be ignored. However, cloud effects on heating can be substantial in the upper troposphere when optical depths are relatively large. Cloud ice also induces a secondary effect on heating in the upper stratosphere due to absorption of reflected shortwave radiation by ozone.

For perturbing aerosols in each layer, we originally planned to use 10% increments in average optical depth at each level as in the illustrations above; however, 10% of the mean values of AAOD (at all levels) and SAOD (at high levels) is smaller than the accuracy limit ($10^{-5}$) in RRTMG. We therefore perturb the optical depth by $10^{-5}$ for those levels. So the $\tau \times 10\%$ in Eq. 2 should be changed to $10^{-5}$ for the situations above.

#### 4.3.2 Longwave radiation

We plot the optical depth at 10 μm in Fig. S5 as representative of aerosol and cloud interactions with longwave radiation. Only the absorption effects of aerosols and cloud ice are considered in RRTMG_LW, as aerosol scattering effects are ignored in this model and cloud ice is treated as a separate aerosol type. Compared with the 550 nm optical depth in Fig. S4, the absorbing aerosol optical depth at 10 μm is smaller by about a factor of two. Decreases in cloud optical depth relative to the shortwave result from both spectral variations and the omission of longwave scattering by cloud ice. Although cloud longwave effects are similar to cloud shortwave effects, the effects of absorbing aerosols on net flux are quite different. Aerosols only induce negative changes in fluxes above the perturbation layer, and the absorbing aerosol effect on longwave heating is about one order of magnitude smaller than that on shortwave heating, affirming the relative weakness of aerosol effects in the longwave part of the spectrum.



## 5 Applications of the UTS aerosol kernels

Kernels serve the purpose of estimating radiative effects at a reduced computational cost in a simple way, as outlined by Huang et al. (2007). When employing kernels, the initial step involves calculating the differences between the target status and reference state, denoted as $\Delta\tau$. Additionally, it is essential to consider either assuming a 200 hPa albedo (or utilizing TOA albedo instead) or determining the tropospheric cloud coverage to select the appropriate kernel set, as detailed in Table 2. Similar to Eq. 3, the aerosol (or cloud) radiative effect can then be calculated as follows:

$$R = \Delta\tau \times k + \overline{R_{\text{ref}}} ,\qquad\qquad(6)$$

If only radiation increments (reductions) are needed, just ignore the last term of the Eq. 6.

With the application method, there are some potential applications for our UTS aerosol kernels, including aerosol effects associated with the ATAL, deep smoke plumes associated with biomass burning, volcanic eruptions, and geoengineering.

### 5.1 ATAL effects

The radiative effects of ATAL aerosols can be reconstructed by combining the AAOD and SAOD radiative kernels derived using the Eq. 3. The kernels further allow for the added context of aerosol longwave radiative effects based only on absorbing aerosol and under different tropospheric scenarios. The results are shown in Fig. 8. To facilitate comparison with the largest aerosol radiative effect of ATAL, the shortwave aerosol optical depth and kernel for 12:00 local solar time in August are selected.

The left panel of Fig. 8 shows AAOD profiles for the zonal-mean reference state (orange line) and in the ASM core region (22.5°N – 25°N, 87.5°E – 90°E, blue line) in August. Below 80 hPa, AAOD concentration is about two times larger in the core region than the zonal mean. Fig. 8 also shows aerosol effects on shortwave and longwave radiative heating in the core region under different tropospheric scenarios. The shortwave aerosol heating rate computed for clear-sky conditions (Gao et al., 2023, Fig. 5) is therefore best compared with the kernel for a 200 hPa albedo of 0.1. Between 110 hPa – 180 hPa, the kernel indicates an aerosol effect of about 0.15 K·day–1, quite similar to that result; however, the shortwave heating effect above 80 hPa is close to 0, smaller than the previous result. This difference may be explained by a combination of the difference in reference states (zonal-mean as opposed to pristine) and radiative transfer models (RRTMG as opposed to libRadtran; Fig. S6). The aerosol effect on longwave heating weakens with decreasing emission temperature and even becomes negative for the smallest 200 hPa emission temperature of 222.9 K. A similar effect has been found for thin cirrus overlying deep convection (Hartmann et al., 2001; Fueglistaler and Fu, 2006). The time-mean 200 hPa albedo in the core region is about 0.24, so that the 200 hPa albedo of 0.25 can roughly represent the all-sky scenario. When considering typical tropospheric cloud cover, the ATAL aerosol effect on shortwave radiative heating in this region is about 0.03 K·day–1 larger than that estimates for clear-sky conditions.



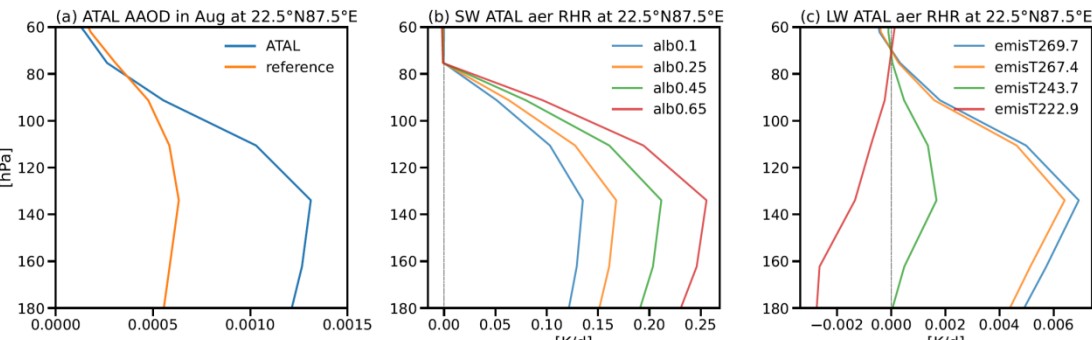

**Figure 8: (a) 550 nm AAOD in the zonal-mean reference profile (orange) and at 22.5°N, 50°E – 120°E (blue) within 180 hPa – 60 hPa, both averaged for August 2011 – 2020. (b) The ATAL aerosol effect on shortwave heating at 22.5°N and 87.5°E with 200 hPa albedo set to 0.1 (blue), 0.25 (orange), 0.45 (green), and 0.65 (red) in August at 12:00 local solar time. (c) The ATAL aerosol effect on longwave heating at the same location with 200 hPa emission temperature set to 269.7 K (blue), 267.4 K (orange), 243.7 K (green), and 222.9 K (red).**

## 5.2 Other potential applications

It is difficult to draw firm conclusions about radiative forcing by stratospheric aerosols associated with extreme events or contingency plans. The uncertain duration and composition, fragmented distribution, complex characteristics, and broad spatial distributions of aerosols during those events increase the difficulty and computational cost of simulating aerosol radiative effects directly using radiative transfer models. Radiative kernels provide a convenient and flexible alternative to estimating these effects, their impacts, and associated uncertainties. This is particularly helpful for the understanding of emergent extreme events (e.g., extreme wildfires).

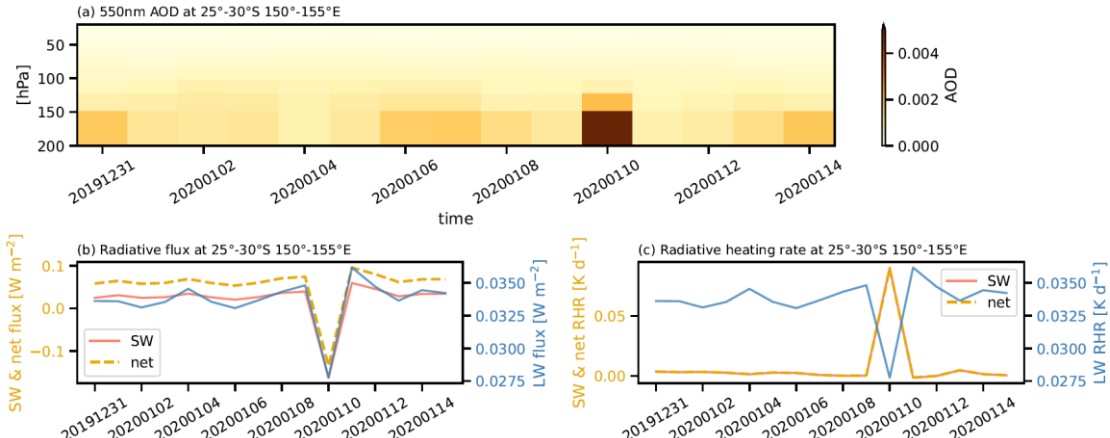

**Figure 9: (a) 550 nm AOD averaged in the 25°S-30°S, 150°E-155°E during 31th Dec. 2019 - 14th Jan. 2020. (b) The radiative flux effect of shortwave (red solid line with y-axis on the left), longwave (blue solid line with y-axis on the**




**right) and net (yellow dashed line with y-axis on the left) around 200 hPa in the 25°S-30°S, 150°E-155°E during 31th Dec. 2019 - 14th Jan. 2020. (c) Same as (b) but for radiative heating rate.**

For example, the large wildfires that occurred in southeast Australia during 2019 – 2020 resulted in a near-tripling of the maximum AOD and substantially larger aerosol concentrations in the stratosphere (Khaykin et al., 2020; van der Velde et al., 2021). In MERRA-2 only the aerosol data from 10th January show this event, on which the 550nm AOD at 200 hPa reached 0.005, plotted in Fig. 9(a). The SW aerosol radiative effect caused by wildfire is about 10 times larger than LW both for radiative flux and radiative heating rate. This wildfire event has caused the radiative flux decrease from 0.1 to -0.1 W m$^{-2}$, but

radiative heating rate increase to 0.1 K d$^{-1}$ near tropopause. Smoke aerosols associated with large and persistent fires can reach high altitudes with large fractions of carbonaceous aerosols (Yang et al., 2021). Volcanic eruptions can also inject aerosols and aerosol precursors directly into the stratosphere. In addition to natural events, certain geoengineering strategies for solar radiation management are based on injecting aerosols or aerosol precursors into the stratosphere. Intended to mitigate global warming, proposals for stratospheric aerosol geoengineering (SAG; MacMartin et al., 2016) include the Stratospheric Particle

Injection for Climate Engineering (SPICE; Pidgeon et al., 2013). The possible side effects of such strategies are not yet fully resolved.

## 6. Summary and discussion

As previous studies have demonstrated the feasibility of simulating aerosol effects on radiative fluxes by using radiative kernels (Thorsen et al., 2020; Kramer et al., 2021), we extend this approach to aerosol effects on radiative heating. Tests indicate that

linear kernels can be used to represent aerosol effects on radiative fluxes for changes in aerosol optical depth (AOD) up to twice the reference state AOD. Aerosol effects on radiative heating can be represented by linear kernels for the full range of tested perturbations, up to at least 10 times the reference state AOD. For the sake of data availability, convenience, and accuracy, we construct kernels based on absorbing AOD (AAOD) and scattering AOD (SAOD). Separate sets of shortwave kernels are constructed for each month and retain diurnal variations at 3-hourly intervals, longwave kernels are independent with time.

To simplify the model, we construct UTS aerosol and cloud ice kernels in latitude bands relative to zonal mean reference states. The lower boundary conditions of the radiative transfer model are set at 200 hPa to roughly represent the state of the underlying troposphere. The radiative impact of the tropospheric state is represented by 200 hPa albedo in shortwave simulations and emission temperature in longwave simulations, with kernels calculated for four scenarios: clear-sky, low cloud, middle cloud, and high cloud. Cirrus clouds in the UTS (p ≤ 200 hPa) are treated as aerosols. Our calculations produce AAOD, SAOD, and

cloud optical depth (COD) shortwave kernels and AOD and COD longwave kernels (RRTMG_LW does not include aerosol scattering effects).

The absorbing aerosol shortwave net flux kernel is positive above the perturbed layer and negative below. As 200 hPa albedo increases, the positive kernel above strengthens and the negative kernel below weakens. The scattering aerosol and cloud



shortwave net flux have no vertical variations because of competing effects on upward and downward fluxes, but their negative
kernels weaken as albedo increases because the effects of scattering saturate. The absorbing aerosol shortwave radiative heating
kernel is much larger than those for scattering aerosol or cloud because the small concentrations of absorbing aerosol are small.
Given the same size of perturbation, the aerosol radiative kernels are stronger for perturbations at higher altitudes.

The aerosol net longwave flux kernels are uniformly negative, with larger values above the perturbation level and smaller
values below. These effects weaken as the 200 hPa emission temperature decreases. Cloud ice kernels likewise show smaller
negative impacts on fluxes below and larger negative impacts on fluxes above. Unlike the shortwave kernels, the longwave
radiative heating kernels are not always positive. Effects on heating weaken when the upward longwave flux from the
troposphere decreases and change sign to local cooling for the smallest upward fluxes (corresponding to a 200 hPa emission
temperature of 222.9 K). As the precision of AOD in RRTMG_SW and RRTMG_LW is 10–5, the small amount of
stratospheric aerosol means that its representation in models is limited by numerical accuracy, an important source of
uncertainty in these and other simulations based on RRTMG.

Inclusion of a clear–clean troposphere in the model simulations results in 200 hPa fluxes that fall within the four scenarios
used for the simulations bounded by 200 hPa. Our results match CERES-based fluxes at 200 hPa well both for longwave and
for shortwave within 9:00 – 15:00 local solar time. Estimated high cloud fractions are larger in CERES than indicated by MLS,
causing our simulations of all-sky 200 hPa downward flux to exceed that based on CERES.

Our tropical kernels are intended to reconstruct the radiative effects of aerosol events in the upper troposphere and stratosphere.
In particular, we show that the kernels can reliably reproduce the ATAL aerosol effects on clear-sky shortwave heating, thus
providing an initial validation of the proposed kernels. These kernels can provide a more flexible and less computationally
expensive means of estimating the radiative effects of the ATAL and other sources of UTS aerosols, including volcanic
eruptions, smoke plumes from large fires, and possible strategies for geoengineering through solar radiation management.

However, many aspects of the tropical stratospheric aerosol radiative kernels remain to be improved. First, large variations in
stratospheric aerosol loading can influence the accuracy of the kernel. For example, the kernels overestimate aerosol effects
on radiative fluxes for values of AOD more than two times larger than the reference AOD. Kernel-based estimates of radiative
heating are much less sensitive to this problem. Second, there are uncertainties associated with our selection of representative
lower boundary conditions, particularly in the distribution of the net longwave flux across individual bands. For example,
whereas a real-world clear-sky scenario would involve a larger effective emission temperature in the window region than in
the $CO_2$ or water vapor bands, we assume a single effective emission temperature across all bands. These limitations will be
addressed in future versions of the kernels.



**Code and Data Availability**

Both MERRA-2 reanalysis data and MLS observation data are available and can be accessed through the NASA Goddard
Earth Science Data and Information Services Center (GES DISC) https://disc.gsfc.nasa.gov/datasets. The ERA5 reanalysis
products are publicly provided by ECMWF https://cds.climate.copernicus.eu/#!/search?text=ERA5&type=dataset. The
CERES data used in this study can be found through the NASA Langley Research Center CERES Data and Information
Archive https://ceres.larc.nasa.gov/data. And RRTMG models are publicly available under their official website
http://rtweb.aer.com/rrtm_frame.html.
The codes for running RRTMG and calculating radiative kernels are available at https://doi.org/10.5281/zenodo.13908975.
All other codes are available from the corresponding author on request.

**Author contributions**

JG: downloaded data, run the model, made validation test, made figures, wrote paper. YH: guided model usage, helped with
technical choices, designed the methodology. JSW: conceived the project, made validation test, helped to revised the paper.
KL: provided advised on kernel application, revised the paper. TG: helped with project development, helped to revised the
paper. QY: helped to design the methodology, provided data, revised the paper.

**Competing interests**

The authors declare that they have no conflict of interest.

**Disclaimer**

Publisher's note: Copernicus Publications remains neutral with regard to jurisdictional claims made in the text, published maps,
institutional affiliations, or any other geographical representation in this paper. While Copernicus Publications makes every
effort to include appropriate place names, the final responsibility lies with the authors.

**Acknowledgements**

We would appreciate Yiran Peng and Ying Zhang for their help on aerosol optical property calculations and RRTMG model
running; each person in Yi Huang's team for the suggestion on radiation; Wenju Cai for his support and constructive comments.





We acknowledge a grant from the Ministry of Science and Technology of the Peoples' Republic of China (Project 2017YFC1501404) and the Natural Sciences and Engineering Research Council of Canada (RGPIN-2019-04511).

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
