# Peer review of "Estimation of aerosol and cloud radiative heating rate in tropical stratosphere using radiative kernel method"

_EGUsphere, 2024_

## Author Comment (AC1)

!     path:        $Source$

!     author:      $Author: miacono $

!     revision:    $Revision: 32234 $

!     created:     $Date: 2018-02-08 15:00:23 -0500 (Thu, 08 Feb 2018) $

!-----------------------------------------------------------------------

!

!-------------------------------------------------------------------------

! Copyright (c) 2002-2016, Atmospheric & Environmental Research, Inc. (AER)

! All rights reserved.

!

! Redistribution and use in source and binary forms, with or without

! modification, are permitted provided that the following conditions are met:

!    * Redistributions of source code must retain the above copyright

!      notice, this list of conditions and the following disclaimer.

!    * Redistributions in binary form must reproduce the above copyright

!      notice, this list of conditions and the following disclaimer in the

!      documentation and/or other materials provided with the distribution.

!    * Neither the name of Atmospheric & Environmental Research, Inc., nor

!      the names of its contributors may be used to endorse or promote products

!      derived from this software without specific prior written permission.

!

! THIS SOFTWARE IS PROVIDED BY THE COPYRIGHT HOLDERS AND CONTRIBUTORS "AS IS"

! AND ANY EXPRESS OR IMPLIED WARRANTIES, INCLUDING, BUT NOT LIMITED TO, THE

! IMPLIED WARRANTIES OF MERCHANTABILITY AND FITNESS FOR A PARTICULAR PURPOSE

! ARE DISCLAIMED. IN NO EVENT SHALL ATMOSPHERIC & ENVIRONMENTAL RESEARCH, INC.,

! BE LIABLE FOR ANY DIRECT, INDIRECT, INCIDENTAL, SPECIAL, EXEMPLARY, OR

! CONSEQUENTIAL DAMAGES (INCLUDING, BUT NOT LIMITED TO, PROCUREMENT OF

! SUBSTITUTE GOODS OR SERVICES; LOSS OF USE, DATA, OR PROFITS; OR BUSINESS

! INTERRUPTION) HOWEVER CAUSED AND ON ANY THEORY OF LIABILITY, WHETHER IN

! CONTRACT, STRICT LIABILITY, OR TORT (INCLUDING NEGLIGENCE OR OTHERWISE)

! ARISING IN ANY WAY OUT OF THE USE OF THIS SOFTWARE, EVEN IF ADVISED OF

! THE POSSIBILITY OF SUCH DAMAGE.

!                 (http://www.rtweb.aer.com/)

!-----------------------------------------------------------------------

USER INSTRUCTIONS FOR RRTMG_SW

Contents:

INPUT_RRTM Instructions
- - - - - - - - - - - - - - - - - - - - - - -

RECORD 1.1

 CXID: 80 characters of user identification (80A1)

  CXID(1) is the flag which determines program initialization and termination. The actual input data stream for RRTM commences with the record containing a '$' in CXID(1). Any records that are read prior to a record containing a '$' in CXID(1) are ignored.

RECORD 1.2

 IAER,  IATM,  ISCAT, ISTRM,  IOUT,  IMCA, ICLD, IDELM, ICOS

  20,   50,   83,   85, 88-90,   94,  95,   99, 100

 18x, I2, 29X, I1, 32X, I1, 1X, I1, 2X, I3, 3X, I1,  I1, 3X, I1,  I1

  IAER  (0,10)  flag for aerosols
     = 0  no layers contain aerosols
     = 6  uses ECMWF global mean aerosol properties for one or all of six aerosol types. Aerosol optical thickness at 0.55 micron, ECAER, must be set manually in the main source module, rrtmg_sw.1col.f90, to activate the aerosols with this option.
     = 10 one or more layers contain aerosols
      (requires the presence of file IN_AER_RRTM)

  IATM  (0,1)  flag for RRTATM  1 = yes

   ISCAT  (0,1) switch for DISORT or simple two-stream scattering
     = 0  DISORT   (unavailable)
     = 1  two-stream (default)

   ISTRM  flag for number of streams used in DISORT  (when ISCAT equal to 0)
     = 0  - 4 streams  (unavailable)
     = 1  - 8 streams  (unavailable)
     = 2  - 16 streams (unavailable)

  IOUT   = -1 if no output is to be printed out.
     =  0 if the only output is for 820-50000 cm-1.

= n (n = 16-29) if the only output is from band n.
For the wavenumbers for each band, see Table I.
= 98 if output is generated for 15 spectral intervals, one
for the full shortwave spectrum (820-50000 cm-1), and one
for each of the 14 bands.

IMCA     (0,1) flag for McICA (Monte Carlo Independent Column Approximation)
for statistical representation of sub-grid cloud fraction
and cloud overlap
= 0    standard forward calculation; do not use McICA (valid for
clear or overcast conditions only)
= 1    use McICA (will perform statistical sample of 200 forward
calculations and output average flux and heating rates)

ICLD     (0,1,2,3) flag for clouds
= 0    no cloudy layers in atmosphere
= 1    one or more cloudy layers present in atmosphere.   Cloud layers
are treated as overcast only for IMCA = 0, or they are treated
using a RANDOM overlap assumption for IMCA = 1.
(requires the presence of file IN_CLD_RRTM for column model)
(available for IMCA = 0 or 1)
= 2    one or more cloudy layers present in atmosphere.   Cloud layers
are treated using a MAXIMUM/RANDOM overlap assmption.
(requires the presence of file IN_CLD_RRTM for column model)
(available only for IMCA = 1)
= 3    one or more cloudy layers present in atmosphere.   Cloud layers
are treated using a MAXIMUM overlap assmption.
(requires the presence of file IN_CLD_RRTM for column model)
(available only for IMCA = 1)

Measurement comparison flags:
IDELM    (0,1) flag for outputting downwelling fluxes computed using the delta-M scaling approximation
= 0    output "true" direct and diffuse downwelling fluxes (default)
= 1    output direct and diffuse downwelling fluxes computed with delta-M approximation
(Note:   The delta-M approximation is always used internally in RRTMG_SW to compute the total
downwelling flux at each level.   What the IDELM flag determines is whether the components
of the downwelling flux, the direct and diffuse fluxes, that are output are the actual direct
and diffuse fluxes (IDELM = 0) or are those computed using the delta-M approximation (IDELM = 1).

ICOS    = 0 there is no need to account for instrumental cosine response (default)

= 1 to account for instrumental cosine response in the computation of the direct and diffuse fluxes

(unavailable)

= 2 to account for instrumental cosine response in the computation of the diffuse fluxes only

(unavailable)

(Note:  ICOS = 1 and ICOS = 2 requires the presence of the file COSINE_RESPONSE, which should

consist of lines containing pairs of numbers (ANG, COSFAC), where COSFAC is the instrumental cosine

response at the angle ANG.)

RECORD 1.2.1

         JULDAT,          SZA, ISOLVAR,      SCON, SOLCYCFRAC, (SOLVAR(IB),IB=16,29)

         13-15,       19-25,     29-30,   31-40,          41-50,     51-190

    12X,    I3, 3X, F7.4,    3X, I2,   F10.4,          F10.5,    14F10.5

         JULDAT         Julian day associated with calculation (1-365/366 starting January 1).
                        Used to calculate Earth distance from sun. A value of 0 (default) indicates
                        no scaling of solar source function using earth-sun distance.

         SZA            Solar zenith angle in degrees (0 degrees is overhead).

         ISOLVAR        Solar variability option [-1,0,1,2,3]

                 =-1 (when SCON .EQ. 0.0): no solar variability; each band uses the Kurucz
                        extraterrestrial solar irradiance, corresponding to a spectrally integrated
                        solar constant of 1368.22 Wm-2 (method used in rrtmg_sw_v3.91 and earlier)
                        =-1 (when SCON .NE. 0.0): solar variability active; baseline solar irradiance
                        of 1368.22 Wm-2 is scaled to SCON, solar variability is determined (optional)
                        by non-zero scale factors for each band defined by SOLVAR

                 = 0 (when SCON .EQ. 0.0): no solar variability; each band uses the solar constant
                        from the NRLSSI2 model of 1360.85 Wm-2 (for the spectral range 100-50000 cm-1)
                        with quiet sun, facular and sunspot contributions fixed to the mean of
                        Solar Cycles 13-24 and averaged over the mean solar cycle
                        = 0 (when SCON .NE. 0.0): no solar variability; baseline solar irradiance of
                        1360.85 Wm-2 (for the spectral range 100-50000 cm-1) is scaled to SCON

                 = 1 solar variability active; solar cycle contribution determined by input of
                        SOLCYCFRAC, a fraction representing the phase of the solar

cycle, with facular brightening and sunspot blocking effects varying in time with this fraction through their mean variations over the average of Solar Cycles 13-24 (corresponding to a solar constant of 1360.85 Wm-2); two amplitude scale factors provided in SOLVAR allow independent adjustment of facular and sunspot effects from their mean solar cycle amplitudes

= 2 solar variability active; solar cycle contribution determined by direct specification of Mg (facular) and SB (sunspot) indices consistent with the NRLSSI2 solar model; these are provided in SOLVAR and are used to model the solar variability at a specific time for a specific solar cycle (SCON = 0.0 only; solar constant depends on Mg and SB indices provided) Further information on setting the Mg and SB indices for this option can be found at the NRLSSI model github site: https://github.com/lasp/nrlssi.

= 3 (when SCON .EQ. 0.0): no solar variability; each band uses the NRLSSI2 extraterrestrial solar irradiance, corresponding to a spectrally integrated solar constant of 1360.85 Wm-2 with quiet sun, facular and sunspot contributions averaged over the mean of Solar Cycles 13-24
= 3 (when SCON .NE. 0.0): solar variability active; baseline solar irradiance of 1360.85 Wm-2 is scaled to SCON, solar variability is determined (optional) by non-zero scale factors for each band defined by SOLVAR and applied to SCON

SCON          For ISOLVAR = -1 or 0:
Total solar irradiance (if SCON > 0, internal solar irradiance is scaled to this value)
For ISOLVAR = 1:
Solar constant; integral of total solar irradiance averaged over solar cycle (if SCON > 0, internal solar irradiance is scaled to this value)

For ISOLVAR = 2:

SCON must be 0.0, since total solar irradiance is defined by the Mg and SB

indices provided in SOLVAR

For ISOLVAR = 3:

Total solar irradiance before individual band scale factors are applied

(if SCON > 0 internal solar irradiance is scaled to this value)

Set SCON = 0.0 to use internal solar irradiance, which depends on ISOLVAR

SOLCYCFRAC    Solar cycle fraction (0-1); fraction of the way through the mean 11-year

cycle with 0.0 defined as the first day of year 1 and 1.0 defined as the

last day of year 11 (ISOLVAR=1 only). Note that for the combined effect of

the solar constant of 1360.85, and the mean facular brightening and sunspot

dimming components (without scaling), the minimum total solar irradiance of

1360.49 occurs at SOLCYCFRAC = 0.0265, and the maximum total solar irradiance

of 1361.34 occurs at SOLCYCFRAC = 0.3826.

SOLVAR        Solar variability scaling factors or indices (ISOLVAR=-1,1,2,3 only)

For ISOLVAR = 1:

SOLVAR(1)    Facular (Mg) index amplitude scale factor
SOLVAR(2)    Sunspot (SB) index amplitude scale factor

For ISOLVAR = 2:

SOLVAR(1)    Facular (Mg) index as defined in the NRLSSI2 model;

used for modeling time-specific solar activity

SOLVAR(2)    Sunspot (SB) index as defined in the NRLSSI2 model;

used for modeling time-specific solar activity

For ISOLVAR = -1 or 3:

SOLVAR(1:14) Band scale factors for modeling spectral variation of

averaged solar cycle in each shortwave band

RECORD 1.4

IEMIS, IREFLECT, (SEMISS(IB),IB=16,29)

          12,          15,                    16-85

  11X, I1,    2X, I1,                   14F5.3

  (Note:   surface reflectance = 1 - surface emissivity)

  IEMIS   = 0 each band has surface emissivity equal to 1.0
          = 1 each band has the same surface emissivity (equal to SEMISS(16))
          = 2 each band has different surface emissivity (for band IB, equal to
SEMISS(IB))

  IREFLECT = 0 for Lambertian reflection at surface, i.e. reflected radiance
      is equal at all angles
          = 1 for specular reflection at surface, i.e. reflected radiance at angle
      is equal to downward surface radiance at same angle multiplied by
      the reflectance.   THIS OPTION CURRENTLY NOT IMPLEMENTED.

  SEMISS     the surface emissivity for each band (see Table I).   All values must be
          greater than 0 and less than or equal to 1.   If IEMIS = 1, only
          the first value of SEMISS (SEMISS(16)) is considered.   If IEMIS = 2
          and no surface emissivity value is given for SEMISS(IB), a value of 1.0
          is used for band IB.

```
* * *
******   these records applicable only if RRTATM not selected (IATM=0)   ******
```

LAYER INPUT    (MOLECULES ONLY)

RECORD 2.1

        IFORM, NLAYRS, NMOL

         2       3-5,    6-10

        1X,I1    I3,     I5

            IFORM     (0,1) column amount format flag
           = 0   read PAVE, WKL(M,L), WBROADL(L) in F10.4, E10.3, E10.3 formats
(default)
           = 1   read PAVE, WKL(M,L), WBROADL(L) in E15.7 format

            NLAYRS     number of layers (maximum of 200)

             NMOL       value of highest molecule number used (default = 7;
maximum of 35)
                   See Table II for molecule numbers.

    RECORD 2.1.1

        PAVE,  TAVE,    PZ(L-1),  TZ(L-1),   PZ(L),  TZ(L)

        1-10, 11-20,     44-51,     52-58,     66-73,  74-80

       F10.4, F10.4, 23X, F8.3,    F7.2,   7X, F8.3,   F7.2

            PAVE    average pressure of layer (millibars) (**If IFORM=1, then PAVE in
E15.7 format**)

            TAVE    average temperature of layer (K)

         PZ(L-1)    pressure at bottom of layer L

         TZ(L-1)    temperature at bottom of layer L  -  used by RRTM for Planck

Function Calculation

             ** NOTE **   PZ(L-1) and TZ(L-1) are only required for the first layer.   RRTM assumes that

                 these quantites are equal to the top of the previous layer for L > 1.

        PZ(L)    pressure at top of layer L

        TZ(L)     temperature at top of layer L  -  used by RRTM for Planck Function Calculation

RECORD 2.1.2

(WKL(M,L), M=1, 7), WBROADL(L)

(8E10.3)

WKL(M,L)    column densities or mixing ratios for 7 molecular species (molecules/cm**2)

WBROADL(L)    column density for broadening gases (molecules/cm**2)

**NOTE** If IFORM=1, then WKL(M,L) and WBROADL(L) are in 8E15.7 format

RECORD 2.1.3    only if (NMOL .GT . 7)         # records depends on NMOL

(WKL(M,L), M=8, NMOL)

(8E10.3)

NMOL is set from LINFIL (TAPE3)

(NMOL limited to 35 in RRTM)    **NOTE: If IFORM=1 then WKL(M,L) in 8E15.7 format**

REPEAT RECORDS 2.1.1 through 2.1.3 for the remaining layers (up to NLAYRS)

```
* * *
********      these records applicable if RRTATM selected (IATM=1)      ********
```

RECORD 3.1

MODEL,   IBMAX,   NOPRNT,   NMOL, IPUNCH,   MUNITS,   RE,      CO2MX, REF_LAT

      5,      15,      25,     30,      35,      39-40, 41-50,      71-80, 81-90

    I5,   5X, I5,   5X, I5,      I5,      I5,     3X, I2, F10.3, 20X, F10.3, F10.3

MODEL      selects atmospheric profile

    = 0   user supplied atmospheric profile
    = 1   tropical model
    = 2   midlatitude summer model
    = 3   midlatitude winter model
    = 4   subarctic summer model
    = 5   subarctic winter model
    = 6   U.S. standard 1976

IBMAX        selects layering for RRTM

    = 0   RRTM layers are generated internally (default)
    > 0   IBMAX is the number of layer boundaries read in on Record 3.3B which are
        used to define the layers used in RRTM calculation

NOPRNT     = 0   full printout
         = 1   selects short printout

NMOL        number of molecular species (default = 7; maximum value is 35)

IPUNCH     = 0   layer data not written (default)
        = 1   layer data written to unit IPU (TAPE7)

MUNITS     = 0   write molecular column amounts to TAPE7 (if IPUNCH = 1, default)

        = 1   write molecular mixing ratios to TAPE7 (if IPUNCH = 1)

RE           radius of earth (km)

         defaults for RE=0:

         a)   MODEL 0,2,3,6     RE = 6371.23 km

   b)        1           RE = 6378.39 km

   c)        4,5       RE = 6356.91 km

CO2MX      mixing ratio for CO2 (ppm).    Default is 330 ppm.

REF_LAT      latitude of location of calculation (degrees)

         defaults for REF_LAT = 0:

         a) MODEL 0,2,3,6     REF_LAT = 45.0 degrees

         b) MODEL 1          REF_LAT = 15.0

         c) MODEL 4,5       REF_LAT = 60.0
* * *
RECORD 3.2

HBOUND,    HTOA

 1-10,   11-20

F10.3,   F10.3

HBOUND        altitude of the surface (km)

HTOA          altitude of the top of the atmosphere (km)

RECORD 3.3   options

RECORD 3.3A         For IBMAX   = 0 (from RECORD 3.1)

AVTRAT, TDIFF1, TDIFF2, ALTD1, ALTD2

 1-10,   11-20,   21-30, 31-40, 41-50

F10.3,   F10.3,   F10.3, F10.3, F10.3

AVTRAT   maximum Voigt width ratio across a layer
          (if zero, default = 1.5)

TDIFF1   maximum layer temperature difference at
          ALTD1 (if zero, default =   5 K)

TDIFF2   maximum layer temperature difference at
          ALTD2 (if zero, default = 8 K)

ALTD1     altitude of TDIFF1 (if zero, default = 0 Km)

ALTD2     altitude of TDIFF2 (if zero, default = 100 Km)

RECORD 3.3B          For IBMAX > 0   (from RECORD 3.1)

              ZBND(I), I=1, IBMAX     altitudes of RRTM layer boundaries

          (8F10.3)

              If IBMAX < 0

      PBND(I), I=1, ABS(IBMAX) pressures of LBLRTM layer boundaries

          (8F10.3)
* * *
  --
* * *
                         User Defined Atmospheric Profile

------------------------------ (MODEL = 0) ---------------------------------

RECORD 3.4

            IMMAX,      HMOD

               5,     6-29

              I5,     3A8

            IMMAX      number of atmospheric profile boundaries

                       If IMMAX is set to a negative value, the level boundaries are
                       specified in PRESSURE (mbars).

            HMOD       24 character description of profile

RECORD 3.5

            ZM,     PM,      TM,      JCHARP, JCHART,    (JCHAR(K),K =1,28)

           1-10, 11-20, 21-30,           36,       37,        41   through   68

          E10.3, E10.3, E10.3,    5x,   A1,       A1,       3X,      28A1

            ZM          boundary altitude (km). If IMMAX < 0, altitude levels are
                        computed from pressure levels PM. If any altitude levels are
                        provided, they are ignored if   IMMAX < 0 (exception: The
                        first input level must have an accompanying ZM for input
                        into the hydrostatic equation)

            PM          pressure (units and input options set by JCHARP)

            TM          temperature (units and input options set by JCHART)

JCHARP        flag for units and input options for pressure (see Table II)

JCHART        flag for units and input options for temperature (see Table II)

JCHAR(K)      flag for units and input options for
              the K'th molecule (see Table II)

RECORD 3.6.1 ... 3.6.N

      VMOL(K), K=1, NMOL

      8E10.3

      VMOL(K) density of the K'th molecule in units set by JCHAR(K)

REPEAT records 3.5 and 3.6.1 to 3.6.N for each of the remaining IMMAX boundaries
* * *
                        User Defined Atmospheric Profile

------------------------------ (IPRFL = 0) --------------------------------

RECORD 3.8

                LAYX,   IZORP,   XTITLE

                  5,       10,     11-60

                 I5,      I5        A50

        LAYX              number of atmospheric profile boundaries

        IZORP (0,1)    flag which determines value of ZORP on Record 3.8.1

                        = 0    ZORP is an altitude in KM
                        = 1    ZORP is a pressure in millibars

        XTITLE          50 character description of profile

RECORD 3.8.1

      ZORP,    (JCHAR(K),K =1,28)

      1-10,      16    through    50

      F10.3, 5X,              35A1

         ZORP          boundary altitude (km) or pressure (millibars) as determined by IZORP
on Record 3.8

        JCHAR(K)       flag for units and input options for
                        the K'th cross-section

                        JCHAR = 1-1              - default to value for specified model
atmosphere

= " ",A              - volume mixing ratio (ppmv)

RECORD 3.8.2 ... 3.8.N

       DENX(K), K=1, IXMOLS

       8E10.3

       DENX(K) density of the K'th cross-section in units set by JCHAR(K)

REPEAT records 3.8.1 to 3.8.N for each of the remaining LAYX boundaries
* * *
TABLE I.   RRTM Bands and Included Species

| Band # | Wavenumber Range (cm-1) | 1050 - 96 mb | 96 - 0.01 mb |
|---|---|---|---|
| 16 | 2600-3250 | $H_2O,CH_4$ | $CH_4$ |
| 17 | 3250-4000 | $H_2O,CO_2$ | $H_2O,CO_2$ |
| 18 | 4000-4650 | $H_2O,CH_4$ | $CH_4$ |
| 18 | 4650-5150 | $H_2O,CO_2$ | $CO_2$ |
| 20 | 5150-6150 | $H_2O,CH_4*$ | $H_2O,CH_4*$ |
| 21 | 6150-7700 | $H_2O,CO_2$ | $H_2O,CO_2$ |
| 22 | 7700-8050 | $H_2O,O_2$ | $O_2$ |
| 23 | 8050-12850 | $H_2O$ | nothing |
| 24 | 12850-16000 | $H_2O,O_2,O_3*$ | $O_2,O_3*$ |
| 25 | 16000-22650 | $H_2O,O_3*$ | $O_3*$ |
| 26 | 22650-29000 | nothing | nothing |
| 20 | 29000-38000 | $O_3$ | $O_3$ |
| 28 | 38000-50000 | $O_3,O_2$ | $O_3,O_2$ |
| 29 | 820-2600 | $H_2O$ | $CO_2$ |

* Included as minor species.

TABLE II. Units and input options for the K'th molecule

TABLE II

USER OPTIONS FOR PRESSURE, TEMPERATURE, AND MOLECULAR DENSITY

JCHARP

PRESSURE        1-6             default to value for specified model atmosphere
(JCHARP)      " ",A          pressure in (mb)
                 B                 "       "    (atm)
                 C                 "         "   (torr)

JCHART

TEMPERATURE       1-6            default to value for specified model atmosphere
    (JCHART)      " ",A         ambient temperature in deg (K)
                   B                 "       "     "     "     "    "   (C)

JCHAR(M)

(M):  AVAILABLE      ( 1)  H2O  ( 2)  CO2  ( 3)    O3 ( 4)    N2O ( 5)    CO ( 6)
CH4 ( 7)    O2
  MOLECULAR  SPECIES      ( 8)    NO   ( 9)  SO2  (10)    NO2 (11)    NH3 (12)  HNO3
(13)    OH (14)    HF
                          (15)  HCL  (16)  HBR  (17)    HI (18)    CLO (19)    OCS (20)
H2CO (21)    HOCL
                          (22)    N2  (23)   HCN  (24) CH3CL (25)   H2O2 (26)  C2H2 (27)
C2H6 (28)    PH3
                          (29) COF2  (30)   SF6   (31)    H2S (32) HCOOH (33) EMPTY (34)
EMPTY (35) EMPTY

potential choice of units for above species:

```
JCHAR = 1-6            - default to value for specified model atmosphere
      = " ",A          - volume mixing ratio (ppmv)
      = B              - number density (cm-3)
      = C              - mass mixing ratio (gm/kg)
      = D              - mass density (gm m-3)
      = E              - partial pressure (mb)
      = F              - dew point temp (K) *H2O only*
      = G              - dew point temp (C) *H2O only*
      = H              - relative humidity (percent) *H2O only*
      = I              - available for user definition
```

JCHAR must be less than "J"
* * *
IN_CLD_RRTM Instructions    (this file required if ICLD = 1 in Record 1.2 of INPUT_RRTM)
* * *
 RECORD C1.1

    INFLAG, ICEFLAG, LIQFLAG

            5          10          15

     4X, I1,   4X, I1,   4X, I1

        Note:    ICEFLAG and LIQFLAG are required only if INFLAG = 2.

            INFLAG = 0 direct specification of optical depths of clouds;
                        cloud fraction and cloud optical depth (gray), single scattering
albedo,
                    and N-str moments of the phase function

                 = 2 calculation of separate ice and liquid cloud optical depths, with
                    parameterizations determined by values of ICEFLAG and LIQFLAG.
                    Cloud fraction, cloud water path, cloud ice fraction, and
                    effective ice radius are input for each cloudy layer for all
                    parameterizations.    If LIQFLAG = 1, effective liquid droplet radius
                    is also needed.    If ICEFLAG = 1, generalized effective size is
                is also needed.

            ICEFLAG = 0 inactive
                     = 1 the optical depths (non-gray) due to ice clouds are computed as
closely as
                        possible to the method in E.E. Ebert and J.A. Curry, JGR, 97, 3831-
3836 (1992).
                     = 2 the optical properties are computed by a method based on the
parameterization
                        of spherical ice particles in the RT code, STREAMER v3.0
(Reference:
                        Key. J., Streamer User's Guide, Cooperative Institute for
                        Meteorological Satellite Studies, 2001, 96 pp.).
                     = 3 the optical depths are computed by a method based on the
parameterization
                    of ice clouds due to Q. Fu, J. Clim., 9, 2058 (1996).

            LIQFLAG = 0 inactive

= 1 the optical depths (non-gray) due to water clouds are computed by a method

based on the parameterization of water clouds due to Y.X. Hu and K. Stamnes,

J. Clim., 6, 728-742 (1993).

These methods are further detailed in the comments in the file 'rrtmg_sw_cldprop.F90'
and the module 'rrtmg_sw_susrtop.F90'.

RECORD C1.2    (one record for each cloudy layer, if INFLAG = 0)

TESTCHAR,      LAY, CLDFRAC,      TAUCLD or CWP,SINGLE-SCAT, PMOM(0:NSTR)
                                ALBEDO
        1,       3-5,      6-15,                    16-25,     26-35,        36-196

        A1, 1X, I3,    E10.5,                    E10.5,     E10.5,       16E10.5

        TESTCHAR     control character -- if equal to '%', cloud input processing
                                is terminated

        LAY          layer number of cloudy layer.   The layer numbering refers to the
                     ordering for the upward radiative transfer, i.e. botton to top.
                     For IATM = 0 (Record 1.2), each layer's number is equal to the
                     position of its Record 2.1.1 in the grouping of these records.
                     For example, the second Record 2.1.1 occurring after Record 2.1
                     corresponds to the second layer.   For IATM = 1 (Record 1.2) and
                     IBMAX > 0 (Record 3.1), layer n corresponds to the region
between
                     altitudes n and n+1 in the list of layer boundaries in Record 3.3B.
                     For IATM = 1 (Record 1.2) and IBMAX = 0 (Record 3.1), the layer
                     numbers can be determined by running RRTM for the cloudless
case
                     and examining the TAPE6 output from this run.

        CLDFRAC      cloud fraction for the layer

        TAUCLD       (INFLAG = 0 only) total (ice and water) optical depth for the layer

        SINGLE-SCATTERING SIngle-scattering albedo for cloudy layer (unitless)
        ALBEDO

        PMOM         Moments of the phase function, from 0 to NSTR. (unitless)

        Note: The true optical depth,single-scattering albedo, and phase function
moments must be input.
        The Delta-M scaling, using the standard Henyey-Greenstein approach, is applied
to the
                     input cloud properties.

RECORD C1.3   (one record for each cloudy layer, INFLAG = 2)

TESTCHAR,      LAY, CLDFRAC,      TAUCLD or CWP, FRACICE, EFFSIZEICE, EFFSIZELIQ

1,      3-5,      6-15,                16-25,      26-35,        36-45,         46-55

A1, 1X, I3,     E10.5,                E10.5,      E10.5,        E10.5,        E10.5

TESTCHAR     control character -- if equal to '%', cloud input processing
             is terminated

LAY          layer number of cloudy layer.   The layer numbering refers to the
             ordering for the upward radiative transfer, i.e. botton to top.
             For IATM = 0 (Record 1.2), each layer's number is equal to the
             position of its Record 2.1.1 in the grouping of these records.
             For example, the second Record 2.1.1 occurring after Record 2.1
             corresponds to the second layer.   For IATM = 1 (Record 1.2) and
             IBMAX > 0 (Record 3.1), layer n corresponds to the region
between

             altitudes n and n+1 in the list of layer boundaries in Record 3.3B.
             For IATM = 1 (Record 1.2) and IBMAX = 0 (Record 3.1), the layer
             numbers can be determined by running RRTM for the cloudless
case

             and examining the TAPE6 output from this run.

CLDFRAC      cloud fraction for the layer.

TAUCLD       (INFLAG = 0 only) total (ice and water) optical depth for the layer
or    CWP    (INFLAG > 0) cloud water path for the layer (g/m2)

FRACICE      (INFLAG = 2) fraction of the layer's cloud water path in the form
             of ice particles

EFFSIZEICE (INFLAG = 2 and ICEFLAG = 1) Effective radius of spherical
             ice crystals with equivalent projected area to hexagonal ice particles
             following Ebert and Curry (1992).
             Valid sizes are 13.0 - 130.0 microns.

             (INFLAG = 2 and ICEFLAG = 2) Effective radius of spherical
             ice crystals, re (see STREAMER manual for definition of this parameter)
             Valid sizes are 5.0 - 131.0 microns.

             (INFLAG = 2 and ICEFLAG = 3) Generalized effective size of

hexagonal
ice crystals, dge (see Q. Fu, 1996, for definition of this parameter)
Valid sizes are 5.0 - 140.0 microns.

NOTE: The size descriptions for effective radius and generalized effective
size are NOT equivalent.   See the particular references for the appropriate
definition.

EFFSIZELIQ (INFLAG = 2 and LIQFLAG = 1) Liquid droplet effective radius, re
(microns)
Valid sizes are 2.5 - 60.0 microns.

IN_AER_RRTM Instructions    (this file required if IAER = 1 in Record 1.2 of INPUT_RRTM)
* * *
RECORD A1.1

        NAER

        3X, I2

        NAER    number of different aerosol types (maximum of 99).   An aerosol type is characterized by a specified
                spectral dependence of aerosol optical depth, single-scattering albedo, and phase function; a change
                to any of these quantities requires a new aerosol type.   Each aerosol type requires the presence of
                Records A2.1 - A2.3.

RECORD A2.1

        NLAY,    IAOD,    ISSA,    IPHA,   (AERPAR(I),I=1,3)

         5,       10,      15,      20,                   21-44

        3X, I2, 4X, I1, 4X, I1, 4X, I1,                  3F8.2

        NLAY    number of layers containing the aerosol with the specified properties: spectral dependence of aerosol
                optical depth (IAOD,AERPAR), single-scattering albedo (ISSA, SSA), and phase function (IPHA,PHASE).
                Note that each layer can contain only one aerosol type.

        IAOD    (0,1)    flag for specifying the spectral dependence of aerosol optical depth
                = 0    spectral dependence determined by Angstrom-like relationship (Molineaux et al.; see below)
                        with variables AERPAR(1), AERPAR(2), and AERPAR(3)
                = 1    aerosol optical depths directly input for each layer and band in Record A2.1.1

ISSA    (0,1)    flag for gray or spectrally dependent single scattering albedo
        = 0        gray SSA (equal to SSA(16))
        = 1        spectrally dependent SSA (for band IB, equal to SSA(IB))

IPHA    (0,1,2)    phase function flag
        = 0        spectrally gray phase function (equal to PHASE(16) in first and only
Record 2.3); uses

                Henyey-Greenstein phase function
        = 1        spectrally dependent phase function (for band IB, equal to PHASE(IB)
in first and only

                Record 2.3); uses Henyey-Greenstein phase function
        = 2        direct specification of moments of phase function.    See Record 2.3.

AERPAR    (only used if IAOD = 0) array of parameters for obtaining aerosol optical
depth as a
                function of wavelength, as described below:

    AOD = AOD1 * (AERPAR(2) + AERPAR(3) * (lambda/lambda1)) /
                    ((AERPAR(2) + AERPAR(3) - 1) + (lambda/lambda1)**AERPAR(1))
            where
                lambda = wavelength in microns
                lambda1 = 1 micron
                AOD = aerosol optical depth at wavelength lambda
                AOD1 = aerosol optical depth at 1 micron (see Record A2.1.1).

        This is a version of Eq. 13 from    Molineaux et al, Appl. Optics, 1998.    The default
values of
        AERPAR(1), AERPAR(2), and AERPAR(3), which are 0, 1, and 0, respectively, yield an
aerosol
        with spectrally grey extinction.

        (Note:    To obtain Angstrom relation, set AERPAR(2)=1., AERPAR(3)=0., and
AERPAR(1) equal to
        Angstrom exponent.)

 RECORD A2.1.1

  (if IAOD = 0)
        LAY,   AOD1

5,    6-12

        2X, I3,    F7.4

    (if IAOD = 1)
            LAY, (AOD(IB),IB=16,29)

                5,                      6-103

        2X, I3,                  14F7.4

            LAY          layer number of aerosol layer.    (The layer numbering refers to the
                         ordering for the upward radiative transfer, i.e. bottom to top.
                         For IATM = 0 (Record 1.2), each layer's number is equal to the
                         position of its Record 2.1.1 in the grouping of these records.
                         For example, the second Record 2.1.1 occurring after Record 2.1
                         corresponds to the second layer.   For IATM = 1 (Record 1.2) and
                         IBMAX > 0 (Record 3.1), layer n corresponds to the region
between
                         altitudes n and n+1 in the list of layer boundaries in Record 3.3B.
                         For IATM = 1 (Record 1.2) and IBMAX = 0 (Record 3.1), the layer
                         numbers can be determined by running RRTM for the cloudless
case
                         and examining the TAPE6 output from this run.(

    (if IAOD = 0)
            AOD1           aerosol optical depth at 1 micron; can be used to scale the
amount of aerosols in the
                  layer; see Record A2.1
    (if IAOD = 1)
            AOD           aerosol optical depth for each band

  REPEAT RECORD A2.1.1 for the remaining layers containing this aerosol type. There should
be NLAY
                         records A2.1.1

    RECORD A2.2

        (SSA(IB),IB=16,29)

                    (1-70)

14F5.2

SSA    Single scattering albedo for each band; must be equal to or greater than zero and

less than or equal to 1.   If ISSA equals 0, then only the first value of SSA (SSA(16))

is considered.

RECORD A2.3

(PHASE(IB),IB=16,29)

(1-70)

14F5.2

PHASE    Moments (starting with first moment) of the phase function for band IB. In this implementation,

the phase function P(u) for each band is defined as

P(u) = sum over streams l { (2l+1) (PHASE_l) (P_l(u)) }

where

u = cos(theta)

PHASE_l = the lth moment of the phase function

P_l(u) = lth Legendre polynomial,

and the number of streams to be used in DISORT (using the delta-M method) is determined

by the value of ISTRM in Record 1.2 of INPUT_RRTM.

For IPHA = 0 or IPHA = 1, the Henyey-Greenstein phase function is used and only the first

moment of the phase function needs to be specified, so only one Record A2.3 is read.

(Note:   The first moment of the phase function is the asymmetry parameter.)   If IPHA equals

0, then only the first value of PHASE (PHASE(16)) is considered.

For IPHA = 2, the number of A2.3 records should be equal to the number of streams.

REPEAT RECORDS A2.1 through A2.3 for the remaining aerosol types.   There should be NAER

sets (A2.1 through A2.3) of records.

---

## Author Response (AR1)

**Juan A. Añel**

> Unfortunately, after checking your manuscript, it has come to our attention that it does not comply with our "Code and Data Policy". https://www.geoscientific-model-development.net/policies/code_and_data_policy.html First, in your work you use the RRTMG models, and in the "Code and Data Availability" section you link its webpage. This webpage is not a suitable repository for scientific publication; therefore, it does not comply with our policy. You must store the RRTMG models that you have used in one of the repositories that we accept. I note here that the RRTMG models do not include a license in their webpage, and as a consequence nobody can use them. It is an usual misunderstanding to think that making code available in a web page makes it free to anyone to use it, which is not the case. I recommend you to communicate with the RRTMG developers to make them aware of this, and include with their code a license that allows you and other to use the model, and deposit it in a suitable repository. Then you must reply to this comment with the link and DOi of the new repository containing the RRTMG models code. You should do this as soon as possible, as in its current version your manuscript does not comply with our policy, and we can not accept in Discussions manuscripts that do not do it. Also, you must include the information on the new repository (DOI an link) in any potentially reviewed version of your manuscript. Moreover, in your manuscript you state " All other codes are available from the corresponding author on request". First, our policy clearly states that all the code necessary to replicate the work exposed in a manuscript must be published and accessible to anyone at submission time. It is not clear what you mean by "all other codes"; however, we can not accept that it is necessary to contact you to get access to such code. Therefore, you must publish it following the same instructions than for the RRTMG models. Please, address this issues and reply to this comment as soon as possible. Otherwise, we will have to reject your manuscript for publication because of lack of compliance with the policy of the journal.

The RRTMG provide a license at the beginning of their instruction file (upload as the supplement file in response), I have uploaded the code of model I used and the instruction with license, at https://doi.org/10.5281/zenodo.14357597. The modifications about code availability in manuscript are in Line 508 - Line 510: "And RRTMG models are available at https://doi.org/10.5281/zenodo.14357597 (official website http://rtweb.aer.com/rrtm_frame.html), the license can be found at the beginning of instruction file. The codes for running RRTMG and calculating radiative kernels are available at https://doi.org/10.5281/zenodo.14359763."

**Hua Zhang**

> This paper estimates the radiative fluxes and heating rates of absorbing aerosols, scattering aerosols, and cloud ice particles in tropical stratosphere using the newly developed radiative kernels. A notable merit of this study is the construction of aerosol kernels, and the application of them has the potential to better understand the radiative effects of aerosols in the upper troposphere and stratosphere.

**Major comments:**

> 1. Line 221. It seems that the kernels are calculated by perturbing aerosols at each level simultaneously. To my knowledge, previous studies all established kernels (e.g., water vapor, cloud) by perturbing the variable at each level at one time. It is necessary to clarify and justify the choice.

Thanks for your comments, I didn't give a clear explaination. I just use kernels perturbing at each level simutaneously when testing kernel method, while the aerosol kernel at each level seperately in application part. To clearify my method, the modifications are: Line 221 "For convenience, aerosols are perturbed by increasing concentrations by 10% at each level simultaneously only when testing the accuracy of various kernel method."

Line 366 "We then use the results of these simulations to construct aerosol and cloud kernels at each level for UTS region." Line 397-Line 400 "R=∑(Δτl×kl)+Rref (6), where l is the atmospheric level. A vertical one-dimensional kernel is calculated for the disturbance of aerosol at each layer, and the total aerosol radiative effect is the sum of that at each layer."

> 2. Line 285. The authors select four reference state boundary conditions for the kernel calculations. It is not clear why these four points are selected and why these points can represent clear-sky, low cloud, middle cloud, and high cloud conditions.

Those four represent points are chosen because of their relatively high frequency in the distribution of 200 hPa shortwave and longwave radiative flux. Due to the significant difference in corresponding albedo (or emission temperature), it could be assume that they represent four different scenarios, and we name these four sceranios as clear sky, low cloud, middle cloud and high cloud.

We modify the pharagraph in Line 283 - Line 288: By analysing joint plots of shortwave and longwave radiative flux at 200 hPa within 30°S – 30°N (only Fig. S3 shown here as 30°S example), we identify four representative points with relatively high frequency between 30°S – 30°N (four red stars in Fig. S3). The albedo and emission temperature calculated from 200 hPa radiative flux are listed in Table 3, which can be regarded as four reference state boundary conditions. Four different albedos could broadly represent four different cloud cover in the underlying troposphere, name as clear-sky, low cloud, middle cloud, and high cloud. The upper troposphere lower stratosphere aerosol kernels are based on these four scenarios, with the frequency decreases sequentially.

> 3. Line 298. It seems that the changes of water vapor and ozone in the upper troposphere and stratosphere are ignored in the simulations. Any estimate of its impact?

We think the change of ozone and water vapor won't influence aerosol radiative effect. To test that, we run the rrtm model with different times of ozone and water vapor, even 1.5 times of them only cause about 0.001 W/m2 difference of aerosol radiative effect for both longwave and shortwave, which can be regarded as systematic error. We can emphasize this in the paper.

Modification in Line 298: Due to the changes of radiatively active components like water vapor and ozone do not significantly affect aerosol radiative effects, they are assumed have no large variations in UTS.

> 4. Line 299. The new kernels are constructed based on several linearity assumptions. However, it is not clear whether the cloud radiative effect varies linearly within a range of COD and whether the total radiative effects of aerosols and cloud ice can be represented as a linear sum of radiative effects associated with AOD and COD. It is necessary to validate these assumptions.

To improve our ability to represent the radiative effects of aerosol-cloud interactions through the kernels, we describe cirrus cloud ice using an aerosol-type input file, in other word, using the same module to calculate aerosol and cloud radiative effect in RRTMG, which is explained in Line 151-154 and shortly mentioned again in Line 301. So the variation of COD has the same pattern as AOD and can be linearly summed. Figure 5, 6 and 7 represent the all-sky radiative flux, which include both aerosol and cloud radiative effects.

To strength this, a sentence is added in Line 220: Since cirrus clouds are regarded as aerosols in model calculation, the radiative effects of cirrus clouds also conform to this conclusion.

> 5. Figures 2, 3, 5-7, S3. The authors select several months (i.e., January, May, July) to validate the assumptions of kernel calculations. Are these months representative? Are the test results in other months consistent with the results in these months?

We have plotted the aerosol and cloud distribution zonally averaged between 30°S – 30°N at each month, the variations across each month are not large, so it doesn't make much difference which month to choose.

I would show the distribution of AAOD, SAOD and COD in supplementary file as Figure S2 - S4, and give an explaination in Line 212: the radiation effects in July is chosen here. Due to the variations in AAOD, SAOD and COD are small in each month (Fig. S2 - S4), consistent conclusions were drawn for the remaining months.

**Other comments:**

> 1. Line 200. Please correct the time range of reference state.

The modification is in Line 200: The atmospheric reference state is taken as the July 2019 average from MERRA-2 and the target state as July 2020.

> 2. Line 314. "Figure 5 and Fig. 6". Please use the uniform expression.

The template of this journal said: "The abbreviation 'Fig.' should be used when it appears in running text and should be followed by a number unless it comes at the beginning of a sentence, e.g.: 'The results are depicted in Fig. 5. Figure 9 reveals that.'". In Line 314, the sentence "Figure 5 and Fig. 6 show comparisons of ..." meets the formatting requirement.

> 3. Lines 413, 420, 478, 496. Please correct the superscript and subscript.

Modification has been done.

> 4. Table 2. Please add RMSE for the kernel calculation.

Modification has been done in Table 2, and some misuse of slope and correlation coefficient in Table 2 have been corrected in this revision.

**Anonymous Referee**

> During the Asian summer monsoon, strong deep convection helps lift aerosols from pollution into the tropical upper troposphere, lower stratosphere where it can impact radiation directly or by changing the thermodynamic and dynamic conditions of those layers. This paper introduces new radiative kernels designed specifically to quantify these components of radiation change in the tropical UTLS where this Asian Tropopause Aerosol Layer (ATAL) occurs. The paper shows these kernels are computationally efficient yet still able to properly represent the total aerosol radiative effects originally simulated directly by fully complex models. This manuscript is well written, and provides a nice approach to constructing aerosol kernels, which is a much needed tool. I provide some minor comments below.

Minor comments:

> 1. Line 48: Matus et al 2019 also developed aerosol kernels that the authors may consider citing
>    and discussing in the into:
>    https://agupubs.onlinelibrary.wiley.com/doi/full/10.1029/2019GL083656

A brief discussion of this paper is in Line 48: Matus et al. (2019) used aerosol radiative kernels based on observations to estimate anthropogenic aerosol radiative forcing. Although their aerosol kernels are calculated by simply dividing changes of aerosols and radiations under linear assumption, the results are reasonable compared with CMIP5.

> 2. Line 98-100: I appreciate the need to use different data sources for different variables, but did
>    the authors evalute these combinations for inconsistencies? For instance, are ERA5 temperatures
>    sufficiently cold for instances where there is nonzero ice water content as reported by MLS? Or is
>    the height reported by ERA5 a reasonable representation of where MERRA-2 thinks the aerosols
>    are located? It would be useful to provide evidence of a sanity check to give the reader
>    confidence that the dataset merging done here was appropriate.

The ERA5 temperatures in recent reanalyses near the tropopause are very similar to each other and in good agreement with radio occultation profiles (Tegtmeier et al., 2020). Heights based on ERA5 and MERRA-2 are somewhat different, mainly because MERRA-2 suffers from a warm bias in the upper troposphere. As a result, geopotential heights in MERRA-2 are somewhat larger than those in ERA5. But since MERRA-2 is just the basic state for the kernel form of Taylor series expansion, so we wouldn't expect this to matter much for the kernel computations. In the other word, we need an aerosol layer just for calculating kernel, but it doesn't need to be fully consistent with the actual situation.

The accuracy of ERA5 is supplemented in Line 100: The temperature from ERA5 is in good agreement with radio occultation profiles (Tegtmeier et al., 2020).

> 3. Line 184-186: What is the shading effect of aerosols? And why is it not impacting the linearity of
>    aerosol radiation at the tropopause? The authors should provide more discussion here about
>    why linearity holds so well at the tropopause calculations.

The discussion of "shading effect" and no longer linear of aerosol radiative effect are in Line 185: That is to say, most of the radiation that can be scattered and absorbed has already been interacted with by the existing aerosol particles, like shading the sunlight. The SW radiation that the newly added aerosol particles can interact with is markedly diminished. As a result, the influence of aerosols on radiation no longer follows a linear relationship.

> 4. Line 213-216: More explanation about why AAOD is so impactful on tropopause radiation and
>    SAOD is not would be helpful here. It may not be intuitive to most readers, who are likely more
>    experienced with TOA or Surface conditions.

The discussion is in Line 219: The significant difference in the impact of absorbing and scattering aerosols on radiative heating rate is that absorbing aerosols can directly absorb radiation and convert the absorbed radiation energy into thermal energy, thereby directly heating the surrounding atmosphere and significantly affecting the radiative heating rate. Scattering aerosols only change the direction of radiation propagation, but do not directly heat the atmosphere, and their contribution to the local radiative heating rate is relatively small.

> 5. Line 258: I can appreciate that aerosol effects have low sensitivity to the background
>    thermodynamic or cloud state? But what about sensitivity to the background aerosol state? I

> would imagine a 10% aerosol perturbation in a high aerosol concentration condition vs a
> pristine condition would lead to a different radiative perturbation. And likewise, the
> heterogeneous spatial pattern of aerosol base state would matter. Is that the case? If so, does it
> mean the kernels are only relevant for simulations where the background aerosol fields are
> similar to those of MERRA2?

The state of the background aerosol is of great significance. However, when using the aerosol radiative kernel, it is not necessary to have the same background. Within the range of linear variation (not exceeding three times to radiative flux and no limitation for radiative heating rate), people can use kernels to calculate a new background radiation with different aerosol concentrations. And then calculate how the perturbation of aerosol could influence radiations with aerosol radiative kernels.

A supplement explanation is added in Line 399: A new reference state can also be defined with difference of aerosol concentrations.

> 6. Line 364: Mention of creating cloud kernels felt quite sudden as there was really no discussion of
>    it in the intro. There are mentions of aerosol-cloud interaction but I thought that was in
>    reference to setting the base state for the radiative transfer calculation. I recommend the authors
>    spend a little more time in the intro or in this section explaining the motivation for making these
>    cloud kernels and why its a natural fit to make these particular cloud ice kernels along with the
>    aerosol kernels

The explaination about why cloud radiative kernel is included is in Line 151: Except aerosols, cloud also plays an important role to influence radiative effect. The cloud above upper troposphere is located at a height similar to aerosols and are mainly composed of ice particles. So the cirrus cloud radiative kernels in upper levels are calculated to compare with aerosols. The establishment of cloud radiative kernel also helps to better simulate radiation changes in the stratosphere. We describe cirrus cloud ice using an aerosol-type input file that specifies optical depth, single scattering albedo, and asymmetry factor.

> Conclusion: Especially since we are heading into CMIP7, I recommend a summary of the types of model
> various, levels and temporal resolutions one would need to apply these kernels to diagnose radiative
> effects. I suspect subdaily data is always needed given the preservation of diurnal information in the
> kernels, and often modeling centers do not provide more than monthly or daily mean data.

There is no limitation for model and temporal resolutions for these kernels, although they don't include hourly data, people can choose the time they want from kernel to calculate corresponding radiative effects. The recommendation of dataset has been added in Line 500: Dataset include 3-dimensional aerosol concentration or cloud optical depth data could use this radiative kernel, choose the corresponding month and local time to estimate stratospheric radiative effects.